# Environmental arginine controls multinuclear giant cell metabolism and formation

Julia S. Brunner [1,2], Loan Vulliard [3], Melanie Hofmann[1,2], Markus Kieler [1,2], Alexander Lercher [3], Andrea Vogel [1,2], Marion Russier[4], Johanna B. Brüggenthies[4], Martina Kerndl [1,2], Victoria Saferding[5], Birgit Niederreiter[5], Alexandra Junza[6,7], Annika Frauenstein[4], Carina Scholtysek[8], Yohei Mikami [9,10], Kristaps Klavins[3], Gerhard Krönke[8], Andreas Bergthaler [3], John J. O'Shea[9], Thomas Weichhart [11], Felix Meissner [4], Josef S. Smolen [5], Paul Cheng[12], Oscar Yanes [6,7], Jörg Menche [3], Peter J. Murray[4], Omar Sharif [1,2], Stephan Blüml [2,5,13]* & Gernot Schabbauer[1,2,13]*

Multinucleated giant cells (MGCs) are implicated in many diseases including schistosomiasis, sarcoidosis and arthritis. MGC generation is energy intensive to enforce membrane fusion and cytoplasmic expansion. Using receptor activator of nuclear factor kappa-B ligand (RANKL) induced osteoclastogenesis to model MGC formation, here we report RANKL cellular programming requires extracellular arginine. Systemic arginine restriction improves outcome in multiple murine arthritis models and its removal induces preosteoclast metabolic quiescence, associated with impaired tricarboxylic acid (TCA) cycle function and metabolite induction. Effects of arginine deprivation on osteoclastogenesis are independent of mTORC1 activity or global transcriptional and translational inhibition. Arginine scarcity also dampens generation of IL-4 induced MGCs. Strikingly, in extracellular arginine absence, both cell types display flexibility as their formation can be restored with select arginine precursors. These data establish how environmental amino acids control the metabolic fate of polykaryons and suggest metabolic ways to manipulate MGC-associated pathologies and bone remodelling.

[1] Institute for Vascular Biology, Centre for Physiology and Pharmacology, Medical University Vienna, 1090 Vienna, Austria. [2] Christian Doppler Laboratory for Arginine Metabolism in Rheumatoid Arthritis and Multiple Sclerosis, 1090 Vienna, Austria. [3] CeMM Research Centre for Molecular Medicine of the Austrian Academy of Sciences, 1090 Vienna, Austria. [4] Max Planck Institute of Biochemistry, 82152 Martinsried, Germany. [5] Division of Rheumatology, Department of Internal Medicine III, Medical University of Vienna, 1090 Vienna, Austria. [6] CIBER of Diabetes and Associated Metabolic Diseases (CIBERDEM), 28029 Madrid, Spain. [7] Metabolomics Platform, IISPV, Department of Electronic Engineering, Universitat Rovira i Virgili, 43204 Tarragona, Spain. [8] Department of Internal Medicine 3, Rheumatology and Immunology, Friedrich-Alexander-University Erlangen-Nürnberg (FAU) and Universitätsklinikum Erlangen, 91054 Erlangen, Germany. [9] Molecular Immunology and Inflammation Branch, NIAMS, National Institutes of Health, Bethesda, MD, Bethesda, MD 20892, USA. [10] Division of Gastroenterology and Hepatology, Department of Internal Medicine, Keio University School of Medicine, Shinanomachi, Shinjuku-ku, Tokyo 160-8582, Japan. [11] Center of Pathobiochemistry and Genetics, Institute of Medical Genetics, Medical University of Vienna, 1090 Vienna, Austria. [12] Bio Cancer Treatment International Ltd., 999077 Hong Kong, China. [13] These authors contributed equally: S. Blüml, G. Schabbauer. *email: stephan.blueml@meduniwien.ac.at; gernot.schabbauer@meduniwien.ac.at

Exacerbated bone resorption by tissue resident osteoclasts underlies the pathogenesis of multiple chronic inflammatory joint diseases, which are significant burdens to human health[1]. Tartrate-resistant acid phosphatase (TRAP)-positive multinucleated osteoclast generation from myeloid progenitors relies on cytokines, most importantly receptor activator of nuclear factor kappa-B ligand (RANKL) and macrophage colony stimulating factor (M-CSF)[2]. However, depending on the tissue, the transcriptional landscape of resident cells is also dictated by their surroundings, with changes in extracellular microenvironment through amino acid (AA) supplementation/restriction regulating immune cell plasticity[3–6]. A hallmark of this plasticity is the fine tuning of cellular metabolism that allows immune cells, especially macrophages to dynamically adjust their metabolism according to available extracellular energy sources. Such metabolic flexibility in response to changing environmental conditions represents an important paradigm in innate immune cell biology[4,6,7]. Analogously, cancer cells are able to adjust their metabolism in the face of nutrient shortage by rewiring their metabolism to efficiently utilize alternative metabolites, thereby sustaining their proliferation[8]. Nonetheless, compensation cannot occur for every nutrient as enzymes that deplete specific AAs can be utilized to limit auxotrophic cancer cell growth. For example, L-asparaginase is used in acute lymphatic leukaemia treatment, but its therapeutic efficacy is limited by side effects[9]. Recent clinical studies employed pegylated recombinant arginase 1 (recArg1 or BCT-100) to deplete arginine in various cancers, yet impacts of arginine availability on other disease settings remains relatively unexplored[10,11]. We reasoned that manipulating environmental arginine could alter local cellular metabolism within bone, possibly through impacts on tissue resident osteoclasts and thus change the trajectory of arthritis-induced bone loss.

To test this hypothesis, we used recArg1 to study the effects of arginine depletion in arthritis and osteoclastogenesis. RecArg1 exerts beneficial effects in murine arthritis by dampening osteoclast metabolism and differentiation. In arginine scarcity, osteoclasts and IL-4-induced multinucleated giant cells (MGCs) display flexibility in their development, as their formation can be restored by supplementation with select arginine precursors. Our results highlight the importance of environmental arginine in MGC development and show therapeutic effects of systemic extracellular arginine depletion in murine arthritis.

## Results

**RecArg1 exerts beneficial effects in murine arthritis.** To investigate if systemic arginine manipulation modulated bone diseases, we first used recArg1 in serum transfer arthritis (K/BxN), which is characterized by RANKL-induced bone erosion (Supplementary Fig. 1a)[12]. Arginine depletion during disease caused improved clinical outcomes as evaluated by increased grip strength associated with reduced paw swelling (total score) and weight loss. Histological analysis revealed decreased formation of osteoclasts in arthritic paws, whereas we found negligible effects of arginine bioavailability on joint inflammation (Fig. 1a). Using a different arthritis model, we found arginine was equally important in tumour necrosis factor (TNF)-dependent arthritis (hTNF$^{Tg/+}$, Supplementary Fig. 1b)[13]. Its restriction improved clinical disease parameters, independent of weight, and led to decreased formation of bone resorbing osteoclasts, suggesting a predominant effect of recArg1 treatment on osteoclast formation in arthritis (Fig. 1b). Finally, we systemically depleted arginine in collagen-induced arthritis (CIA), a model also dependent on adaptive immunity (Supplementary Fig. 1c)[14]. In vivo imaging revealed attenuated osteoclast-specific paw cathepsin K (Ctsk) amounts during arginine restriction, accompanied by decreased

clinical scores, unconnected from weight gain (Fig. 1c). Local Ctsk, Acp5 and Tnf transcripts increased in paws of ill versus healthy animals, confirming that the disease was linked to elevated osteoclastogenesis and inflammation. Although arginine depletion decreased Ctsk and Acp5 mRNA expression, no effects were observed on Tnf mRNA, indicating that arginine depletion had a preferential effect on the transcriptional program of osteoclastogenesis rather than a global effect on inflammation (Supplementary Fig. 1d). Concordantly, while a secondary collagen challenge ex vivo induced robust splenocyte proliferation in diseased versus healthy animals, we detected no difference in recArg1-treated versus sham-treated arthritic animals (Supplementary Fig. 1e). Systemic arginine amounts in healthy animals were 200–300 μM, slightly above adult humans[15] and as expected, depleted by recArg1 during disease (Fig. 1d). This was associated with increases in ornithine and other AAs (Supplementary Fig. 1f). Of note, in CIA, myeloid populations including osteoclast precursors were unaffected by arginine restriction, as was the systemic RANKL/OPG ratio in all models studied (Supplementary Fig. 1g–i), suggesting that decreased osteoclast numbers found in arthritis were due to differentiation of osteoclasts. Collectively, we concluded that recArg1 exerts beneficial effects in murine arthritis, likely by dampening osteoclastogenesis.

To validate a potential cross-talk between the arginine pathway and bone degrading osteoclasts in patients suffering from rheumatoid arthritis (RA), we evaluated patients with respect to their osteoclast activity, grouping them into those suffering from erosive and non-erosive RA. Serum crosslaps, collagen degradation fragments produced from bone resorbing osteoclasts, confirmed their enhanced activity in erosive versus non-erosive patients (Fig. 1e). Versus healthy controls, arginine serum amounts exhibited a tendency to be enhanced in humans with erosive RA (Fig. 1f). In these patients, arginase 1 (ARG1) was significantly increased versus non-erosive patients and positively correlated with patient erosion scores (Fig. 1g). Together these data suggest of a physiological role of ARG1 in limiting osteoclast activity by arginine removal and illustrate an importance of the arginine pathway as well for human bone degradation in RA.

**Extracellular arginine removal attenuates osteoclastogenesis.** To investigate the molecular and cellular mechanisms associated with arginine restriction in bone disease, we next dissected osteoclastogenesis induced with M-CSF/RANKL in the presence or absence of recArg1 using different systems-type approaches. For simplicity, we refer to M-CSF + RANKL treatment as RANKL and M-CSF + RANKL + recArg1 treatment as recArg1, unless otherwise indicated (Fig. 2a). The control RANKL transcriptome consisted of 464 mRNA expression changes with 249 being down- and 215 upregulated. As expected, the latter included hallmark osteoclastic genes; Fos, Nfatc1, Acp5 and Jdp2[16] and KEGG analysis showed enrichment for osteoclast differentiation (Supplementary Fig. 2a, b). At the mRNA level, perturbations in arginine biosynthesis and arginine and proline metabolism suggested an osteoclast-dependent importance for arginine metabolism (Fig. 2b). Indeed, recArg1 completely blocked murine osteoclastogenesis, an effect abolished by enzyme denaturation and independent of effects on cell viability and also inhibited human osteoclastogenesis (Fig. 2c–f). Treatment with recArg1 affected RANKL-induced gene expression already after 24 h, preceding RANKL effects on osteoclast fusion that mainly occurred at 72–96 h (Fig. 2e, Supplementary Figs. 2c and 3a), indicating recArg1 affects osteoclastogenesis by selectively reducing RANKL-dependent gene expression. These data suggest an important link between RANKL signalling and arginine metabolism and that depletion of extracellular arginine upon recArg1

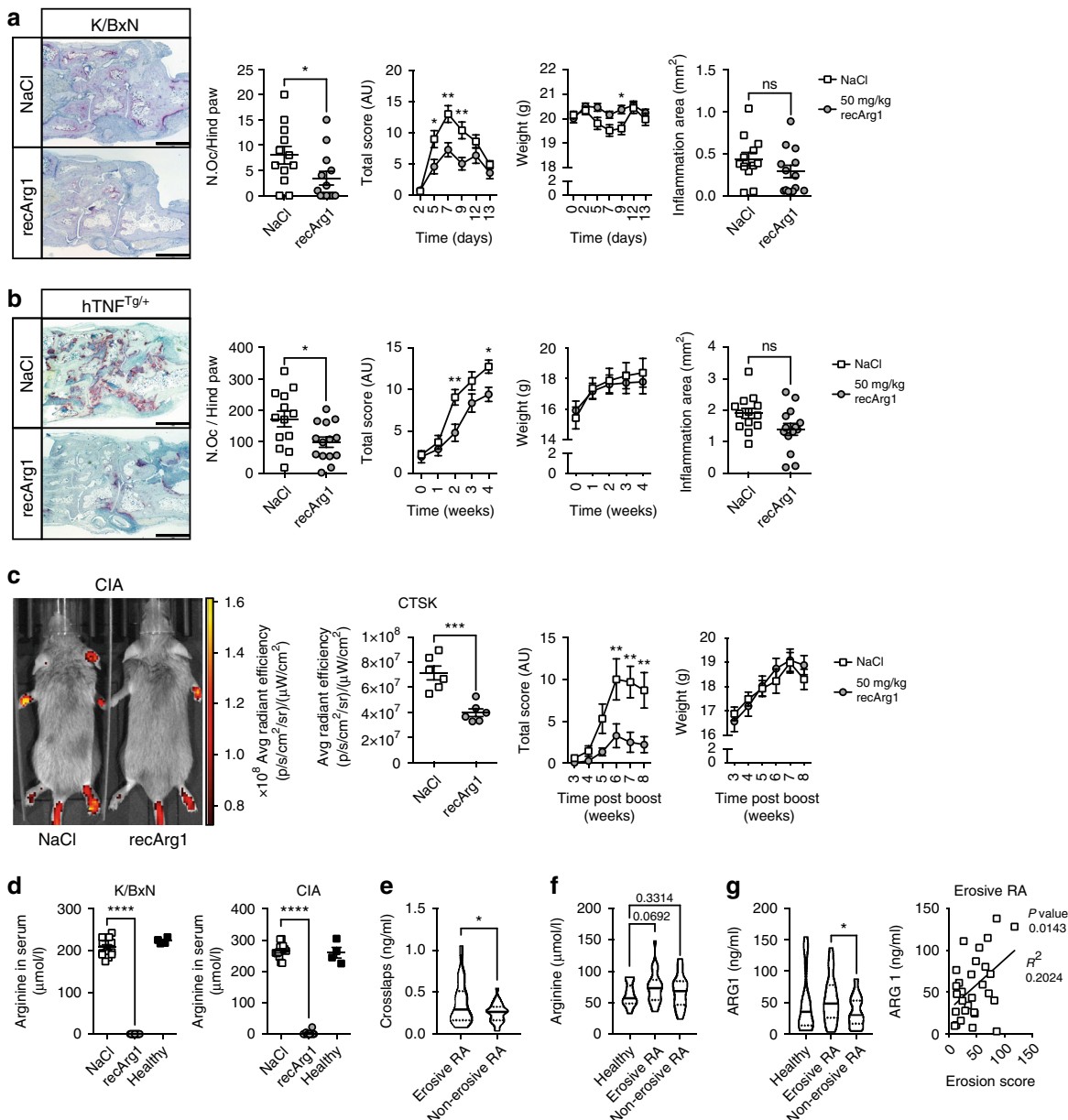

**Fig. 1 Recombinant arginase 1 (recArg1) improves outcome in diverse murine arthritis models and arginase 1 is elevated in erosive RA patients.**
**a**, **b** Paw histology, osteoclast numbers per hind paw (N. Oc), total scores, weight and histology inflammation area of mice suffering from serum transfer arthritis (**a** K/BxN, NaCl n = 12, recArg1 n = 13 animals) or the hTNF$^{tg/+}$ mouse model (**b** NaCl n = 13, recArg1 n = 14 animals). Scale bar represents 1 mm. **c** Ctsk IVIS and quantification, total scores and weights of mice suffering from collagen-induced arthritis (CIA, NaCl n = 13, recArg1 n = 14 animals, Ctsk IVIS n = 6 animals). **d** RecArg1 depletes serum arginine; K/BxN (n = 13 animals per group), CIA (NaCl n = 13, recArg1 n = 14 animals). NaCl in **a**–**d** represents saline vehicle control group. **e** Serum crosslaps in the indicated patient groups (erosive RA n = 30, non-erosive RA n = 29 patients). **f** Serum arginine levels of all patient groups (healthy n = 19, erosive RA n = 29, non-erosive RA n = 30 patients). **g** Arginase 1 levels in all patient groups and correlation between arginase 1 and erosion score of patients suffering from erosive RA (healthy n = 19, erosive RA n = 29, non-erosive RA n = 30 patients). Data are mean ± SEM, *P < 0.05, **P < 0.01, ***P < 0.001, ****P < 0.0001, unpaired t-test (**a**–**c**, **e**–**g**), one-way ANOVA (**d**) and two-way ANOVA post-hoc pairwise comparisons with Bonferroni correction (**a**–**c**), linear regression (**g**). Source data are provided as a Source Data file.

treatment results in dynamic transcriptional changes that attenuate osteoclastogenesis.

To evaluate the importance of intracellular arginine degradation by cellular ARG1 during RANKL-induced osteoclast differentiation, we used the Tie2-cre system to selectively delete *Arg1* in hematopoietic osteoclast precursors[17]. Deficiency of cellular ARG1 within myeloid precursors did not affect osteoclast differentiation, indicating environmental decreases in extracellular arginine mediated by recArg1 exhibit distinct functions with respect to osteoclast differentiation versus those mediated by

cellular ARG1 (Fig. 2g–h). Underscoring the negligible effects observed of conditional *Arg1* deletion on osteoclastogenesis, we observed persistent downregulation of *Arg1* transcript and protein levels post RANKL treatment during osteoclastogenesis of wildtype preosteoclasts (Fig. 2h–i, Supplementary Fig. 2a). To further test the specificity of extracellular arginine in RANKL signalling and osteoclastogenesis, we next treated innate immune cells with recArg1 during their differentiation in an identical manner to osteoclasts. RecArg1 exerted minor influences on macrophage and dendritic cell differentiation (Supplementary

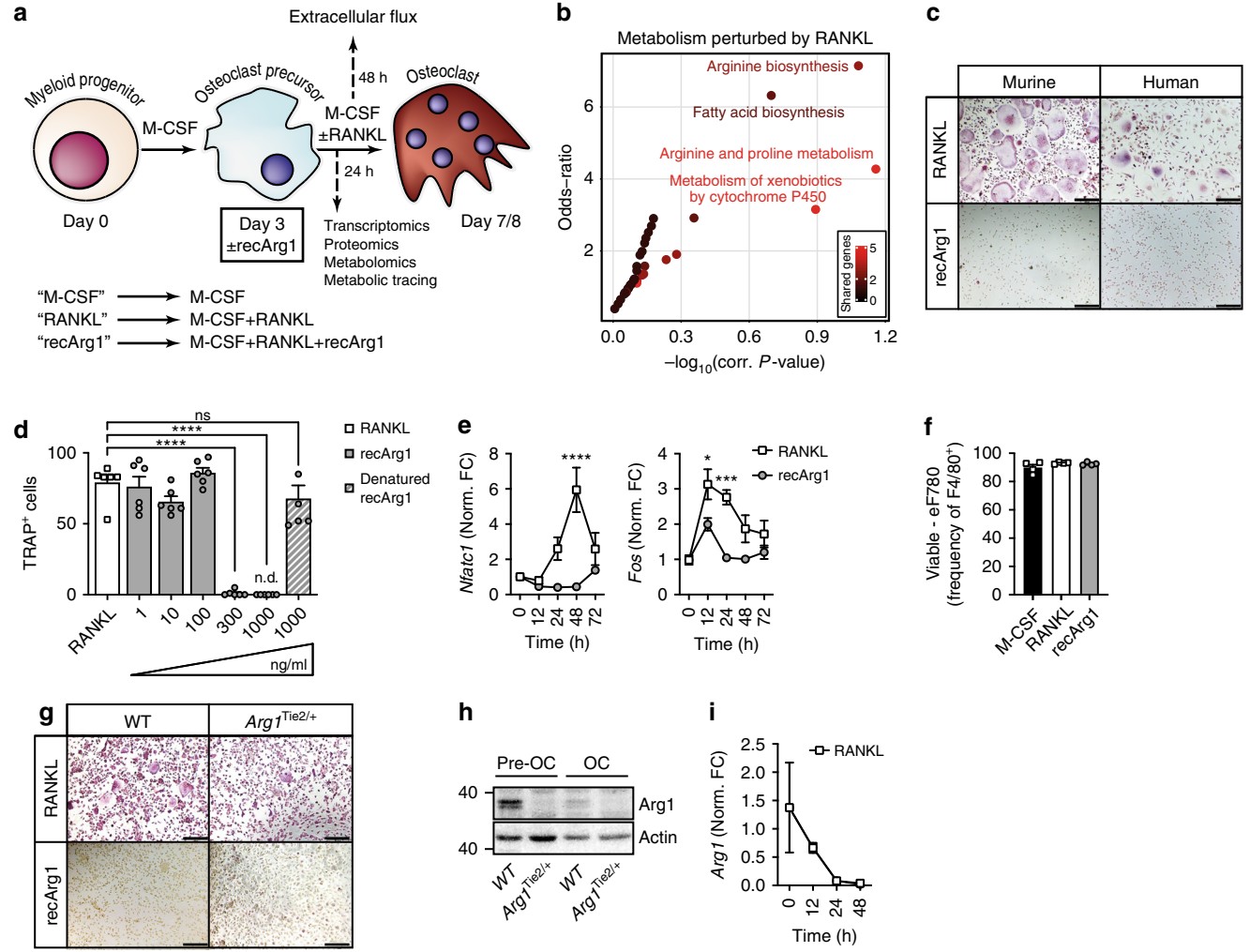

**Fig. 2 Extracellular arginine is essential for RANKL-induced osteoclastogenesis. a** Schematic of systems-wide approaches performed to understand the effect of recArg1 on RANKL dependent osteoclastogenesis. **b** KEGG metabolic enrichment of differentially expressed genes upon RANKL treatment. **c** Representative TRAP stainings depicting that recArg1 abolishes murine and human RANKL-induced osteoclastogenesis. **d** Quantification of murine data depicted in **c** ($n = 6$). **e** qRT-PCR time course of *Nfatc1* and *Fos* ($n = 3$). **f** Quantification of viable cells 24 h post RANKL/recArg1 treatment ($n = 4$). **g**, **h** Osteoclastogenesis is unchanged by *Arg1* deficiency. TRAP stainings (**g**) and Western blots of ARG1 in preosteoclasts (Pre-OC) and osteoclasts (OC) (**h**). **i** qRT-PCR time course of *Arg1* ($n = 3$). Data are mean ± SEM, *$P < 0.05$, ***$P < 0.001$, ****$P < 0.0001$, one-way ANOVA (**d**) and two-way ANOVA post-hoc pairwise comparisons with Bonferroni correction (**e**). Scale bar represents 200 μm (**c**, **g**). Source data are provided as a Source Data file.

Fig. 4), suggesting differential environmental arginine requirements for M-CSF/GM-CSF versus RANKL signalling, the latter of which is well described to be important for multinucleated osteoclast formation.

**RecArg1 counteracts RANKL cellular programmes**. Up to now, we utilized exogenous recArg1 to deplete extracellular arginine to block osteoclastogenesis. We next determined the relative contributions of selective arginine depletion versus other potential effects of recArg1 including ornithine and urea generation, which are the products of the arginase reaction[6]. We compared preosteoclasts treated with RANKL (RANKL) versus recArg1 (RANKL/Arg-Depletion) or cultured in Arg-Free media (RANKL/Arg-Starvation). As controls, we re-supplemented arginine into Arg-Free media (RANKL/Arg-Rescue) and included M-CSF-treated preosteoclasts (M-CSF) starved of arginine (M-CSF/Arg-Starvation) (Fig. 3a). Plotting similarity between transcriptomic samples using multi-dimensional scaling (MDS), we found Arg-Starvation altered RANKL-induced effects, while the M-CSF transcriptome was

minimally affected, confirming arginine requirements for RANKL signalling. RANKL/Arg-Depletion demonstrated more pronounced segregation from RANKL than RANKL/Arg-Starvation samples, whilst both profiles were distinct from M-CSF conditions (Fig. 3b and Supplementary Fig. 5a). Nonetheless, when we plotted RANKL-dependent gene expression in Arg-Depleted versus Arg-Starved cells, we observed highly correlated signatures (Fig. 3c). While recArg1 had the strongest effect on the RANKL transcriptome, most genes (119) significantly modified by Arg-Starvation (150) were shared with Arg-Depletion (420), although genes specific for both were identified (Supplementary Fig. 5b). Thus, we concluded that recArg1 largely acts on RANKL-induced MGC generation via arginine depletion, rather than via the products of the Arg1 reaction.

To dissect the cellular and metabolic changes caused by arginine restriction during osteoclast development, we next used a proteomic approach to probe whether Arg-Depletion in preosteoclasts caused an altered proteome (Fig. 3a, Supplementary Fig. 5c). We did not observe a global translational decrease upon Arg-Starvation or Arg-Depletion, and indeed, many

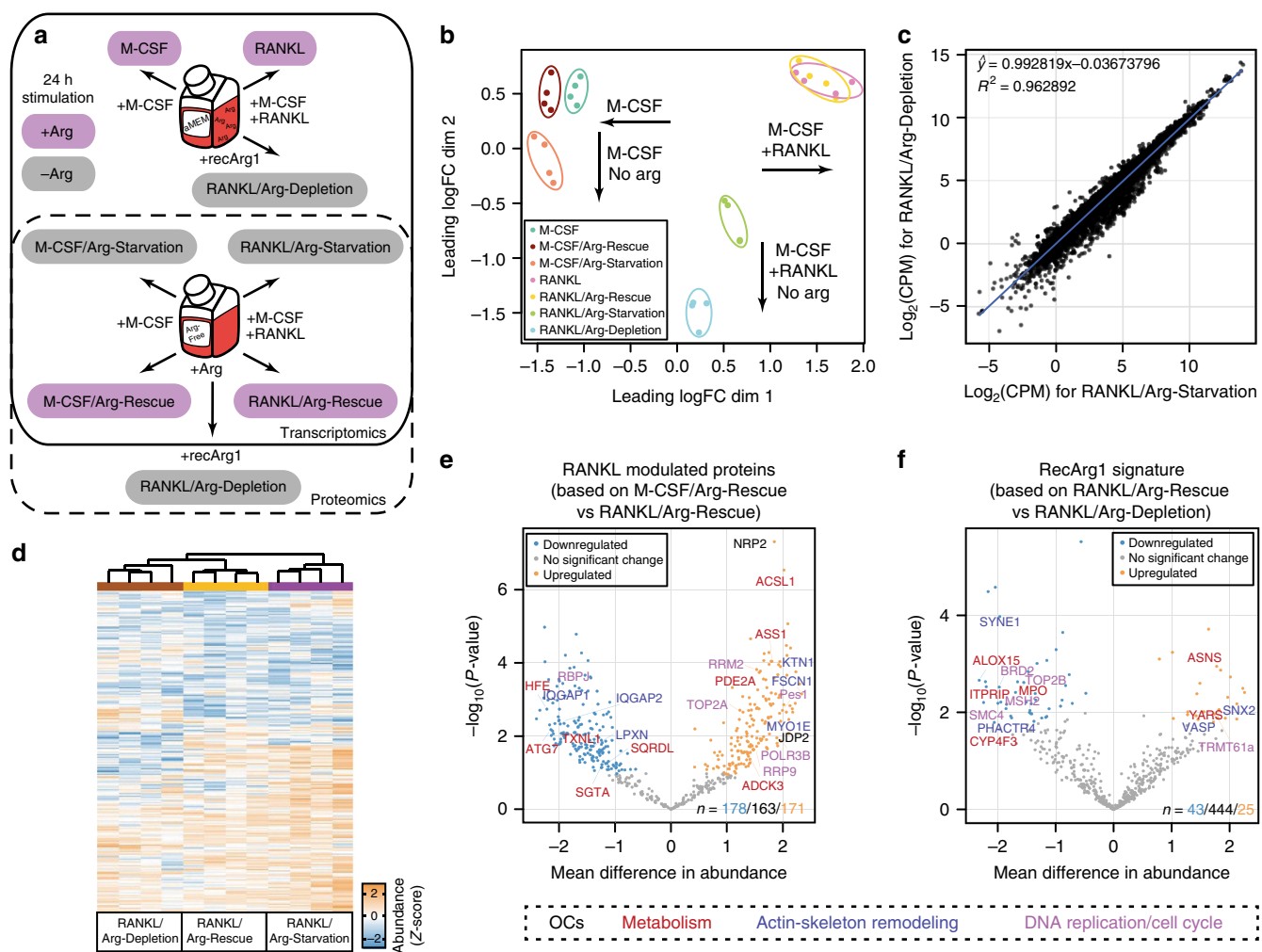

**Fig. 3 Arginine presence specifically sustains RANKL gene and protein expression. a** Workflow dissecting recArg1 specificity on RANKL signalling. Arg-sufficient (pink) and Arg-deficient (grey) conditions in arginine sufficient (aMEM) or deficient media (Arg-Free) depicted. **b** MDS of transcriptomic datasets in **a**, keeping all genes expressed in at least three of the samples. **c** Linear regression of average gene expression ($n = 4$) for RANKL/Arg-Depletion against RANKL/Arg-Starvation. **d** Heatmap of proteomics data in **a**, showing Z-score of abundance level per protein across all conditions (RANKL-conditions shown only, full heatmap in Supplementary Fig. 5d). **e** Volcano plot showing difference between M-CSF/Arg-Rescue versus RANKL/Arg-Rescue ($n = 4$). Highlighted are proteins modulated by RANKL in an arginine-dependent manner. **f** Volcano plot showing difference between RANKL/Arg-Rescue versus RANKL/Arg-Depletion ($n = 4$). Highlighted are proteins uniquely influenced by recArg1.

proteins were induced under low arginine conditions (Fig. 3d, Supplementary Fig. 5d). RANKL addition controlled expression of 349 out of 512 identified proteins with increases/decreases in 171/178 proteins, respectively (Fig. 3e). Only 43 RANKL-induced proteins were counteracted by recArg1 treatment, including those associated with osteoclastogenesis as we expected, as well as proteins involved in the actin cytoskeleton, DNA replication and the cell cycle (Fig. 3f). Thus, as found for the transcriptome, Arg-Depletion had a selective effect on proteins regulated by the RANKL pathway. Examining cell cycle status revealed that while arginine was required for RANKL-induced G$_2$/M cell cycle progression that preceded cell fusion, its withdrawal was equivalent to M-CSF- treated cells, suggesting diminished cellular proliferation (Supplementary Fig. 3a, b). Importantly, although arginine depletion through recArg1 modulated unique proteins, many mediated similar biological processes to the RANKL signature (Fig. 3f). Taken together, these data allowed us to conclude that RANKL cellular programming relies on extracellular arginine, as recArg1 largely counteracted the RANKL-dependent transcriptome and proteome.

**Osteoclast oxidative metabolism is arginine dependent**. Arginine withdrawal has often been associated with inhibition of mTORC1 signalling, a key regulator of cellular proliferation and translation[18]. Concordant with minor effects of arginine deprivation on preosteoclast translation, M-CSF/RANKL mediated mTOR phosphorylation and activation of downstream signalling pathways (pS6K, 4E-BP1) but was largely unaltered by recArg1 (Fig. 4a). In addition, genetic deletion of mTOR signalling components, *Tsc2* and *Rictor*[19], an arginine sensor important for mTORC1 responses (*Slc38a9*)[18,20] or components of the integrated stress response including the known AA sensor GCN2 (*Eif2ak4*) and one of its downstream transcriptional mediators (*Ddit3*)[21], could not bypass the effects of arginine withdrawal on osteoclastogenesis (Fig. 4b).

Having excluded roles of key arginine sensing pathways, we next probed the effect of arginine on osteoclast metabolism. Integrative network analysis of arginine effects on the RANKL signature, unifying transcriptomic and proteomic data, suggested that oxidative metabolism represented an important metabolic hub linked to arginine presence (Supplementary Fig. 6a and Supplementary Table 3). Indeed, we observed RANKL-induced

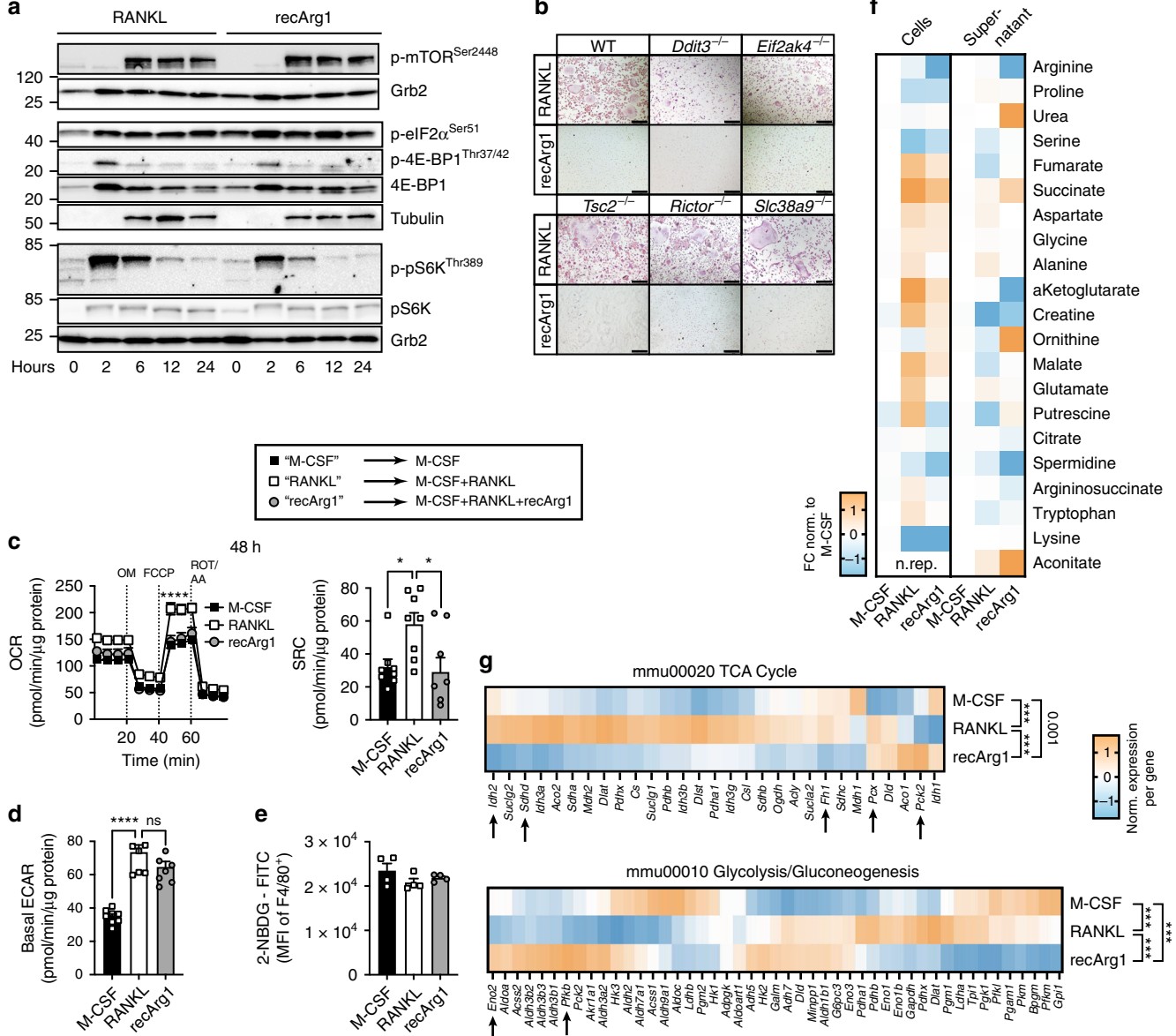

**Fig. 4 Dynamic changes in metabolism to decreased extracellular arginine during RANKL-dependent osteoclastogenesis occur mTOR independently. a** RecArg1 exerts negligible effects on RANKL-mediated mTOR-related pathways. **b** Deficiency in mTOR/AA-starvation components cannot rescue effects of Arg-Depletion. Scale bar represents 200 μm. **c–e** Oxygen consumption rate (OCR), spare respiratory capacity (SRC) and basal glycolytic rate (ECAR) of preosteoclasts ± recArg1 (**c, d** $n = 8$). Negligible effects of arginine deprivation on preosteoclast 2-NBDG (glucose) uptake (**e** $n = 4$). All data 48 h post stimulation. **f** RANKL differentially changed metabolites intra-/extracellularly ± recArg1. Data normalized against M-CSF (left) and represent mean ($n = 4$, RANKL intracellular $n = 3$; see Supplementary Fig. 7c, d) **g** Transcriptional profiles of selected KEGG pathways. TCA cycle and serine/purine biosynthetic enzymes indicated. Data represent mean ($n = 4$). Significance levels according to empirical permutation test, empirical $P$-values reported. Data are mean ± SEM, $*P < 0.05$, $****P < 0.0001$, one-way ANOVA post-hoc pairwise comparisons with Bonferroni correction (**c–e**) and two-way ANOVA post Sidak's multiple comparisons test (**c**). Source data are provided as a Source Data file.

bursts in preosteoclast spare respiratory capacity at 48 h post stimulation, as previously described[22]. These RANKL-mediated increases were counteracted by recArg1, in a manner that was independent of changes in mitochondrial mass (Fig. 4c and Supplementary Fig. 7a, b)[23]. Moreover, versus the M-CSF control, RANKL markedly induced preosteoclast glycolytic rates, independent of changes in glucose uptake (Fig. 4d, e). Elevated preosteoclast glycolysis under RANKL treatment was independent of arginine as no decrease was observed with recArg1 addition (M-CSF+RANKL+recArg1) (Fig. 4d, e). Using a metabolomic approach, we observed RANKL-dependent induction of key intracellular tricarboxylic acid (TCA) cycle

intermediates (e.g. malate, fumarate, succinate, α-ketoglutarate) 24 h post stimulation and only subtle changes in intracellular AAs (e.g. arginine, glycine, alanine) (Fig. 4f, Supplementary Fig. 7c). Oxygen consumption rates (OCRs) 24 h post RANKL treatment were unchanged compared to M-CSF, indicating that RANKL-induced changes in TCA metabolites preceded the observed boost in OXPHOS (Supplementary Fig. 7e).

Arginine is an integral part of the urea cycle, which is connected to the TCA cycle via a shunt through fumarate, linking both cycles[24]. RANKL-induced TCA metabolites were in agreement with robust TCA cycle enzymatic transcription, e.g. those converting isocitrate to α-ketoglutarate (Idh2), succinate to fumarate (Sdhd)

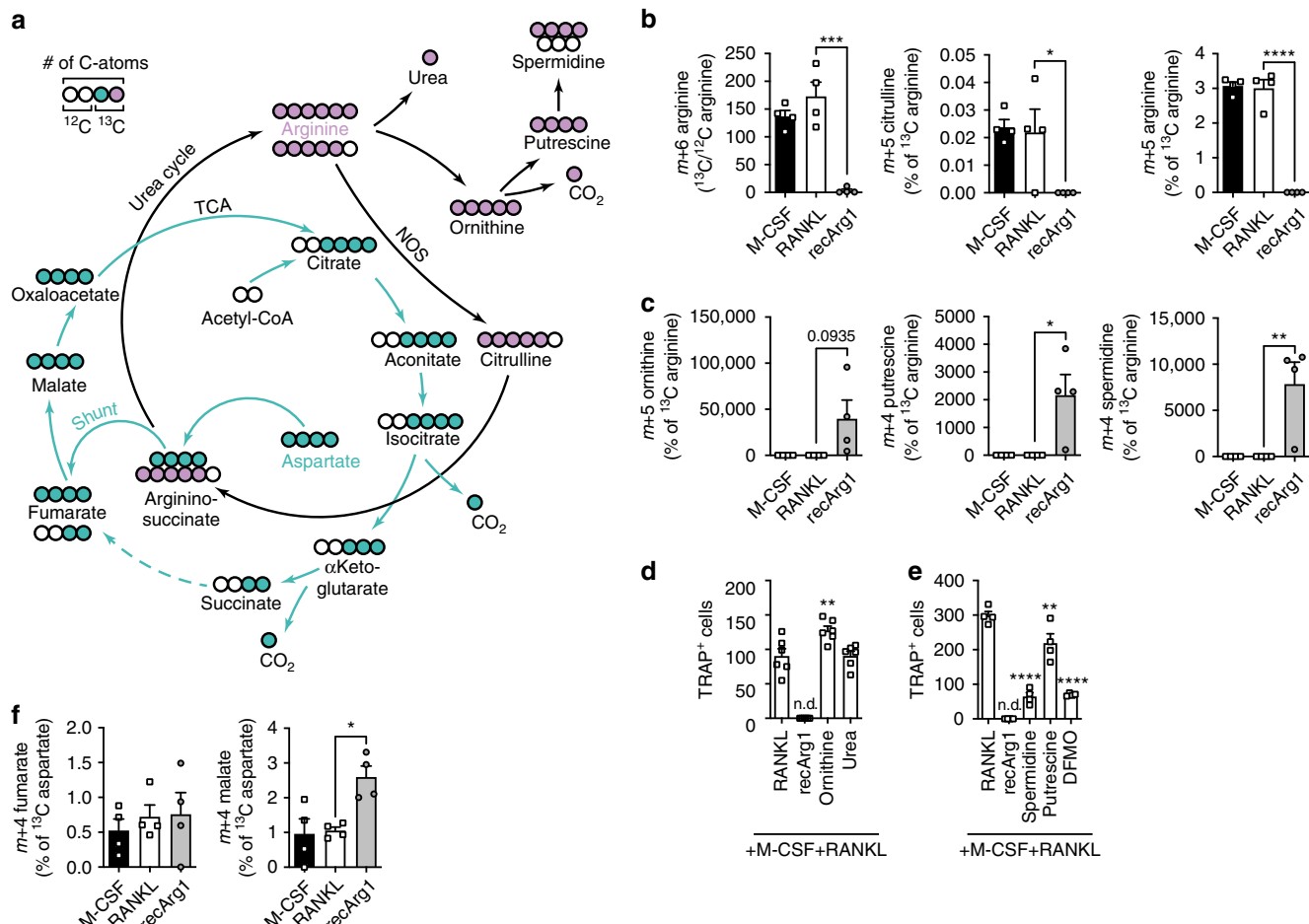

**Fig. 5 Metabolic tracing of arginine and aspartate reveal arginine withdrawal instigates a dysregulated TCA cycle. a** Schematic of the fate of the $^{13}$C-labelled atoms deriving from arginine (pink) or aspartate (turquoise). **b** Ratio of intracellular $^{13}C_6$-labelled arginine to unlabelled $^{12}C_6$ arginine pool and percent $m + 5$ citrulline and $m + 5$ arginine normalized to the labelled $^{13}C_6$ arginine pool ($n = 4$). **c** Intracellular $m + 5$ ornithine, $m + 4$ putrescine and $m + 4$ spermidine normalized to $^{13}C_6$-labelled arginine input ($n = 4$). **d** Excess of recArg1 degradation products ornithine and urea do not impact osteoclastogenesis ($n = 6$). **e** Effect of extracellular polyamines and polyamine synthesis inhibitor difluoromethylornithine (DFMO) on osteoclastogenesis ($n = 4$). **f** Intracellular $m + 4$ fumarate and $m + 4$ malate normalized to $^{13}C_4$-labelled aspartate input ($n = 4$). Data are mean ± SEM, *$P < 0.05$, **$P < 0.01$, ***$P < 0.001$, ****$P < 0.0001$, one-way ANOVA post-hoc pairwise comparisons with Bonferroni correction (**b**–**f**). Source data are provided as a Source Data file.

and fumarate to malate (Fh1)[24] (Fig. 4g). RecArg1 further led to extracellular arginine depletion via conversion to ornithine and urea and reversed the intracellular build-up of malate, fumarate, succinate and α-ketoglutarate induced by RANKL (Fig. 4f and Supplementary Fig. 7c, d), suggesting arginine depletion was linked to a rewired TCA cycle. Indeed, recArg1 counteracted specific RANKL-induced TCA enzyme expression, while upregulating serine and purine biosynthetic enzymes (Pck2, Eno2, Phgdh) (Fig. 4g, Supplementary Fig. 2f)[8]. Consistent with the notion of diverted biosynthetic fluxes upon arginine depletion, increased expression of the pyruvate anabolic genes Glud1 and Gpt2 occurred (Supplementary Fig. 2d). Collectively, our data show that extracellular arginine is required for successful RANKL-dependent TCA cycle upregulation and consecutive increases in oxidative phosphorylation. Our data further suggest that either arginine turnover impacts the TCA cycle, or that the increases in OXPHOS by RANKL are linked to arginine-dependent transcription of TCA cycle enzymes and related metabolite induction or a combination of both.

**RecArg1 instigates a dysregulated TCA cycle in osteoclasts**. To distinguish between the possibilities noted above, we next

performed $^{13}$C-isotopologue labelling experiments. As the carbon backbone of arginine cannot serve as a direct source for TCA cycle intermediates[25], we supplied $^{13}C_6$ arginine in combination with $^{13}C_4$ aspartate in the first 24 h of osteoclastogenesis and measured intracellular isotopologue distribution via liquid chromatography–mass spectrometry (LC–MS) analysis. In this way, we could assess the contribution of each carbon source to both the urea and the TCA cycle simultaneously, as aspartate can be readily converted through argininosuccinate into fumarate via the TCA-Urea cycle shunt (Fig. 5a). $^{13}C_6$ arginine was readily taken up by both M-CSF- and RANKL-treated cells, as we detected 100–200 times more intracellular labelled arginine in relation to unlabelled arginine. In line with negligible increases in arginine transporter transcription (Slc7a1, Slc7a2) (Supplementary Fig. 2e)[26], we observed no difference in $^{13}C_6$-labelled arginine uptake upon RANKL treatment post 24 h. Both M-CSF and RANKL treated cells engaged in arginine recycling as indicated by $m + 5$ labelling in both citrulline and arginine (Fig. 5b). Notably, as arginine was readily converted to ornithine and urea by recArg1 in the supernatant, we could barely detect labelled intracellular $^{13}C_6$ arginine upon recArg1 treatment (Figs. 4f, 5b, Supplementary Fig. 7d). Relative to the low amounts of

intracellular $^{13}C_6$ arginine, under arginine depletion, we detected high quantities of $^{13}C_5$-labelled ornithine and its downstream metabolites putrescine and spermidine, because the enhanced ornithine generated by recArg1 was further catabolized to form polyamines and came from the extracellular $^{13}C_6$ arginine supplied (Fig. 5c). Suggestive of uptake and/or synthesis from $m+5$ ornithine occurring under $^{13}C$ labelling, total intracellular amounts of the polyamines putrescine and spermidine were decreased upon recArg1 treatment in comparison to RANKL (Figs. 4f, 5c, Supplementary Fig. 7c). RNA sequencing analysis demonstrated that similar to macrophages, ornithine transcarbamylase (OTC) was not expressed in preosteoclasts[27] (see deposited data), therefore ornithine cannot be converted to citrulline in these cells. Together, these data indicate that recArg1 treatment converts all extracellular arginine to ornithine and the polyamines as demonstrated by the lack of $m+5$ citrulline and $m+5$ arginine (Fig. 5b, c).

Excess of extracellular ornithine and urea did not have any consequences on osteoclastogenesis, proving that the recArg1-instigated blocks in osteoclastogenesis were not due to increases in these metabolites (Fig. 5d). It was previously shown that extracellular polyamines negatively regulate osteoclastogenesis[28]. However, although recArg1 led to a shift in putrescine and spermidine amounts, their addition at excess (500 µM putrescine and 10 µM spermidine) during osteoclastogenesis caused an approximately 30 and 75% decrease in TRAP-positive cells, respectively. Thus, although accumulation of polyamines upon recArg1 might negatively impact osteoclastogenesis, their increased synthesis cannot fully account for the complete absence of TRAP-positive cells observed under arginine deprivation (Fig. 5e). As we wanted to test if inhibitory effects of recArg1 treatment on osteoclastogenesis were mediated by decreased intracellular polyamine synthesis (Fig. 4f, Supplementary Fig. 7c), we incubated cells with alpha-difluoromethylornithine (DFMO), thereby blocking conversion of ornithine to putrescine[29]. Despite DFMO reducing osteoclastogenesis, multinucleated mature osteoclasts were still observed (Fig. 5e). Overall, these data suggest an importance for polyamines in the context of osteoclastogenesis. However, the effects of recArg1 cannot be solely accounted by deregulated polyamine metabolism.

To prove an operating urea–TCA cycle shunt in preosteoclasts, we simultaneously performed a $^{13}C_4$ aspartate tracing. Suggesting the aspartate–argininosuccinate–fumarate shunt functioned in preosteoclasts independent of arginine presence, we observed equal $m+4$ fumarate labelling in all conditions (Fig. 5f). Notably, as fumarate was $m+4$ labelled, we concluded the carbon source stemmed from argininosuccinate, as direct conversion from aspartate to oxaloacetate would have yielded $m+2$ fumarate (Fig. 5a). Consistent with arginine deprivation globally down-regulating TCA cycle enzymes (Fig. 4g), we observed higher abundance of $m+4$ malate relative to $^{13}C_4$ aspartate input following recArg1 treatment compared to M-CSF and RANKL control conditions (Fig. 5f). Although arginine and its catabolic products cannot directly enter the TCA cycle, the metabolic tracing data utilizing fully labelled aspartate strengthen the notion that arginine presence is required for a functioning TCA cycle.

**AA precursors compensate for AA absence in osteoclastogenesis.** So far we showed that the arginine-dependent dysregulated TCA cycle-initiated cell cycle arrest was not associated with known properties of quiescence including decreased glycolysis, reduced translation rates or mTORC1 deactivation (Figs. 3, 4 and Supplementary Fig. 3b). This suggested a discordant/distinct type of preosteoclast metabolic quiescence, different from that of T cells or hematopoietic stem cells[30–32]. We noted an in-between

state of arginine-restricted preosteoclasts between the M-CSF and the M-CSF/RANKL signature (Fig. 3b), raising the possibility that arginine depletion resulted in a 'poised' preosteoclast state that was associated with a transient reversible osteoclastogenesis-block. Indeed, osteoclast formation could be restarted even after prolonged arginine starvation by arginine resupplementation and even exhibited a tendency towards being increased relative to controls (Supplementary Fig. 3c, d). Notably, the Arg-Starvation-mediated inhibition of polykaryons was restored by arginino-succinate and citrulline supplementation, but not by ornithine and proline, indicating that select arginine precursor metabolites bypass the dependence of arginine for RANKL signalling (Fig. 6a). α-Ketoglutarate was not able to rescue arginine absence (Supplementary Fig. 3e), suggesting that arginine scarcity could not be compensated by direct TCA cycle refuelling but only by AA derivatives. Together, dynamic metabolic adaptation to environmental arginine deficiency sustains cellular fitness until intermediates are re-supplied and RANKL-dependent cellular differentiation programs continued. In line with this finding, in vivo recArg1 administration hindered but did not completely abolish osteoclastogenesis, suggesting that arginine intermediates can compensate for arginine scarcity in vivo (Fig. 1).

To evaluate whether other AAs were essential for RANKL cellular programming, we next cultured preosteoclasts in media selectively devoid of AAs and demonstrated commonalities in metabolic blockage during osteoclastogenesis to a nutrient-limited environment, especially regarding essential AAs (Fig. 6b, Supplementary Fig. 8, Supplementary Table 1). As α-ketoiso-caproate, ketoisoleucine and phenylpyruvate are intermediates of the osteoclastogenesis essential AAs leucine, isoleucine and phenylalanine, respectively, and can enter the TCA cycle eventually via acetyl-CoA[33], we next evaluated their compensatory potential for the aforementioned AAs. In line with TCA/urea cycle intermediates citrulline and argininosuccinate compensating for arginine deficiency (Fig. 6a), non-proteogenic metabolites α-ketoisocaproate, ketoisoleucine and phenylpyruvate could rescue defects in osteoclastogenesis that occurred in the absence of their related AA (Fig. 6c). Therefore, these data suggest that osteoclasts can adjust to the absence of environmental AAs by utilizing their respective precursors or intermediates for successful osteoclastogenesis.

**IL-4 MGCs and osteoclasts share arginine requirements.** Noting that mononuclear innate immune cell in vitro differentiation efficiency is independent of arginine absence (Supplementary Fig. 4) and that RANKL induces giant cell formation, we next examined whether the concept of arginine reliance holds true for other MGCs. IL-4 induces myeloid cell multinucleation[34], resulting in MGCs that are distinct from osteoclasts but share common characteristics[35]. Strikingly, extracellular arginine removal using recArg1 prevented IL-4-generated MGCs (Fig. 7a). Identical to the unremarkable requirements of intracellular ARG1 deficiency for RANKL-induced osteoclastogenesis, IL-4 robustly induced MGCs in the absence of $Arg1$ (Fig. 7b). Using RT-PCR, we confirmed the previously observed effect of recArg1 on the expression of select RANKL-modulated metabolic enzymes $Pck2$ and $Sdhd$ (Figs. 5g, 7c). RANKL- or IL-4-dependent regulation of $Sdhd$ was further associated with increased expression of the MGC signature gene $Dcstamp$. Notably, recArg1 treatment exhibited identical effects on both enzymes in both cell types, inducing $Pck2$ and downregulating $Sdhd$ (Fig. 7c). Further, arginine starvation abolished IL-4-induced MGC formation and this was compensated by the same urea cycle intermediates in MGCs as in osteoclasts (Figs. 6a and 7d, e) and associated with improved OXPHOS (Fig. 7f).

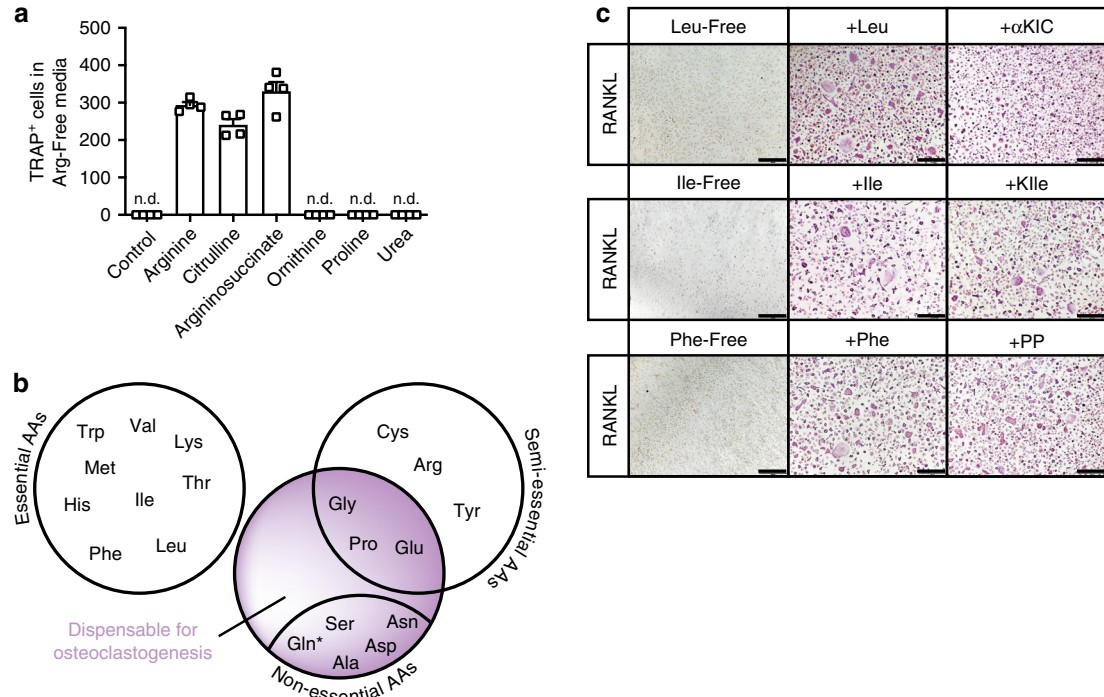

**Fig. 6 Intermediates or precursors of amino acids including arginine can compensate for amino acid absence in osteoclastogenesis. a** Urea cycle metabolites can/cannot rescue arginine requirements in osteoclastogenesis in Arg-Free media ($n = 4$). **b** Schematic demonstrating which amino acids are essential (white) versus non-essential (pink) for osteoclastogenesis. Star relates to TRAP-positivity with decreased occurrence of MGCs ($n = 4$). **c** Immediate intermediates of leucine, isoleucine and phenylalanine degradation compensate for the lack of these amino acids and rescue osteoclastogenesis. Representative TRAP stainings are shown ($n = 4$). aKIC alpha-ketoisocaproate, Ile isoleucine, KIle ketoisoleucine, Leu leucine, Phe phenylalanine, PP phenylpyruvate. Scale bar represents 200 µm. Data are mean ± SEM. Source data are provided as a Source Data file.

## Discussion

Formation of MGCs such as osteoclasts is complex with previous studies identifying a range of essential molecules, ranging from surface receptors such as RANK to co-stimulatory molecules and transcription factors[2,16,35]. Here, we demonstrate that in addition to those factors, the extracellular availability of the AA arginine is critical for RANKL and IL-4 induced differentiation of myeloid precursor cells. While previous studies highlighted an importance for cellular ARG1 in negatively regulating osteoclastogenesis[36], here we demonstrate that extracellular arginine is essential for RANKL-induced metabolism. We found extracellular arginine presence is critical for RANKL to elicit cellular programs and metabolic changes, especially related to cellular respiration. Arginine withdrawal during RANKL signalling was intrinsically linked to downregulation of TCA cycle enzymes and metabolites. Metabolic tracing experiments further confirmed that a dysregulated TCA cycle under arginine withdrawal occurred, associated with malate accumulation derived from labelled aspartate. These changes occurred early during osteoclastogenesis and preceded arginine-dependent dampening of RANKL-induced bursts in cellular respiration. Together, we demonstrate that in preosteoclasts extracellular arginine presence is required for efficient TCA function in response to RANKL stimulation. Akin to osteoclasts, IL-4 induces the formation of MGCs[34]. These cell types displayed clear similarities in arginine-dependent oxidative metabolism, strengthening the notion of arginine requirements to meet MGC energy demands. Equivalent to osteoclasts, IL-4 MGCs were able to form by utilizing arginine precursors to adapt to arginine scarcity. In contrast to MGC differentiation, the presence of arginine was not required for the terminal differentiation of other myeloid cells, such as bone-marrow-derived macrophages or dendritic cells, suggesting specificity for polykaryon formation.

Interestingly, mTOR is a central metabolic integrator, whose role in osteoclastogenesis remains controversial[37,38]. Importantly, although numerous studies in transformed cells suggested absolute extracellular arginine requirements for mTOR activation[18], our data indicate that arginine is not required to sustain mTOR signalling during RANKL-dependent osteoclastogenesis. Indeed, upon arginine deprivation, we did not observe a reduction in translation rates or diminished glycolysis, but rather a type of preosteoclast metabolic quiescence associated with cell cycle arrest and distinct to that previously described in other cell types such as hematopoietic stem cells[31]. The effects of arginine deprivation on osteoclastogenesis were fully reversible upon arginine resupplementation and independent of increases in recArg1 degradation products, such as ornithine and urea. Notably, arginine absence could be compensated by its precursors citrulline and argininosuccinate, suggesting requirements of a functioning arginine recycling machinery during osteoclast development.

Although our data demonstrate the crucial role of extracellular arginine in osteoclastogenesis, absence of other AAs phenocopied the blocks observed under arginine absence. Thus, osteoclasts most likely couple extracellular AA amounts to their developmental program, as scarcity in single essential AAs could be compensated by osteoclast intrinsic re-synthesis through supplementation of its derivatives or precursors. While here we established TCA cycle dysregulation upon arginine withdrawal, we cannot be certain if this is true for the remaining osteoclastogenesis essential AAs identified in this study. Nonetheless, given osteoclastogenesis is an energy demanding process, it is plausible that similar mechanisms of energy shifts under AA starvation are in place as observed under arginine withdrawal. A plausible model for the effects of essential AAs on the selective metabolic

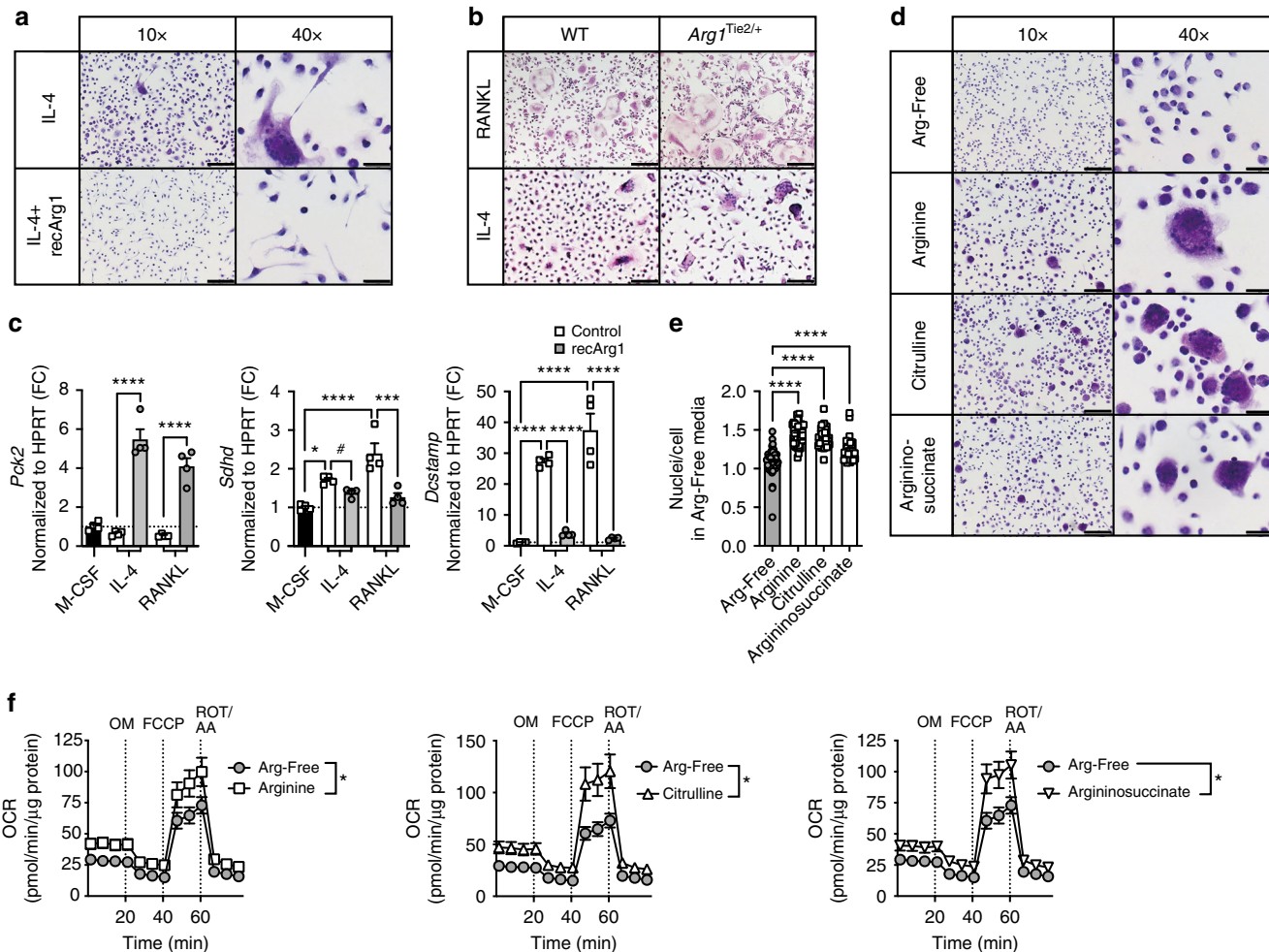

**Fig. 7 Osteoclasts and IL-4-induced multinucleated giant cells (MGCs) share mechanisms of metabolic adaptation in nutrient scarcity.**
**a** Representative H&E stainings of MGCs ± recArg1. **b** Representative H&E stainings of MGCs from *Arg1*[Tie2/+] and WT littermate controls. **c** qRT-PCR of *Pck2*, *Sdhd* and *Dcstamp* in IL-4- (M-CSF + IL-4) and RANKL (M-CSF + RANKL)-induced MGCs ± recArg1. All data correspond to day 7, # corresponds to *P* = 0.01 assessed by *t*-test (*n* = 4). **d**, **e** Representative H&E stainings (**d**) and quantifications (**e**) of nuclei per cell of IL-4-generated MGCs in the absence of arginine (Arg-Free Media), re-supplemented with arginine, citrulline or argininosuccinate. Data represent cells counted in 50 random frames (*n* = 50). **f** IL-4-induced MGC oxygen consumption rate (OCR) in the absence of arginine (Arg-Free Media), re-supplemented with arginine, citrulline and argininosuccinate. Significance was calculated by unpaired *t*-test between area under the curve (AUC) of indicated conditions (Arg-Free *n* = 10, arginine *n* = 12, citrulline *n* = 11, argininosuccinate *n* = 11). Data are mean ± SEM, *$P < 0.05$, **$P < 0.01$, ***$P < 0.001$, ****$P < 0.0001$, one-way ANOVA Tukey post-hoc test and *t*-test (**c**), one-way ANOVA post-hoc pairwise comparisons with Bonferroni correction (**e**) and two-way ANOVA post Sidak's multiple comparisons test (**f**). Scale bar represents 200 μm (10×), 50 μm (40×) (**a**, **d**) and 200 μm (RANKL) and 100 μm (IL-4) (**b**). Source data are provided as a Source Data file.

'poised' state of osteoclastogenesis is that osteoclast progenitors 'integrate' the lack of a key AA(s), which then triggers a pathway that causes a selective block in the TCA cycle. For example, AA-starved osteoclast progenitors could induce the downregulation or degradation of key central metabolic enzymes, linking the pathways. In this model, the effects of low AAs would be rapidly reversible upon AA restoration and osteoclast development could proceed, as we observed. Several examples of such reversible metabolite signalling are known, including oxygen and iron sensing[39,40]. Clearly, such an 'information transfer system' would need to be independent from the mTOR and GCN2 pathways as we found. Nevertheless, a 'sensor' must integrate information from all the AAs described herein that block osteoclastogenesis, or individual sensors for each AA may exist. However, such a model needs to also account for the 'rescue' effect of each AA precursor such as citrulline. Ultimately, we propose that AA presence is required to sustain oxidative metabolism, which is essential for multinucleation, and that this process may involve

yet to be discovered metabolic regulatory pathways. Together, our data establish how availability of environmental AAs, especially arginine, control polykaryon developmental programs and metabolism. They further imply therapeutic strategies for arginine depletion in MGC-mediated diseases.

## Methods

**Human studies**. Human patient serum was used freshly for osteoclastogenesis assays or processed and stored until analysis according to standard operating procedures by the MedUni Wien Biobank, a central facility included in a certified quality management system[41]. All study subjects provided informed consent. This study was approved by the local ethics committee of the Medical University Vienna (EK #559/2005). Arginine and ARG1 levels were assessed using ELISA (Arginine: Immundiagnostik AG #K7733; human ARG1: Antibody Online Gmbh #ABIN1113579).

**Animals and arthritis models**. Female wildtype animals (C57BL/6J, RRID: IMSR_JAX:000664 and DBA/1J, RRID: IMSR_JAX:000670) used for in vivo arthritis models were purchased from Charles River Laboratories. Serum-transfer arthritis was induced by intraperitoneal application of 150 μl of K/BxN serum on

day 0 and day 2 in C57BL/6J and animals were harvested post 13 days treatment[42]. Human tumour necrosis factor (hTNF[Tg/+], RRID: MGI:3053718) transgenic animals were identified from tail tissue PCR using hTNFTg primers (5′-TACCC CCTCCTTCAGACACC-3′ and 5′-GCCCTTCATAATATCCCCCA-3′) and were used for experiments at 5–6 weeks of age. Mice were harvested after 5 weeks of treatment. Female wildtypes (DBA/1J) were immunized subcutaneously with 50 μg of chicken type II collagen (Sigma-Aldrich #C9301) in 50 μl of H₂O, emulsified in 50 μl of Freund's complete adjuvant (Sigma-Aldrich #F5881) that was enriched with 10 μg/ml *Mycobacterium tuberculosis* (Difco/BD Biosciences #H37Ra), on day 1 and day 21. Mice were harvested after 8 weeks of treatment. Treatment regimen comprised a dose of 50 mg/kg recArg1 or NaCl administered intra-peritoneally twice weekly, starting 1 day before disease induction or at 5–6 weeks of age regarding hTNF[Tg/+] mice. hTNF[Tg/+] mice were a kind gift of the George Kollias[13]. Arginine levels were assessed using metabolomics or via ELISA (see below). Mouse TNFSF11/RANKL (Boster #EK0843) and mouse TNFRSF11B/osteoprotegerin (abcam #ab203365) serum levels were measured using ELISA. All animal procedures were approved by the local ethics committee of the Medical University Vienna (BMWFW-66.009/0013-V/3b/2019 and BMWFW-66.009/0227-WF/V/3b/2017) and were conducted in strict accordance with Austrian law.

**Clinical assessment of murine arthritis.** Clinical signs of arthritis are described by a well-established semi-quantitative double-blind score system[43]. Swelling per paw was recorded: 0 to 3 (0 = no swelling; 1 = mild swelling of toes/ankle; 2 = moderate swelling of toes/ankle; 3 = severe swelling of toes/ankle). Grip strength per paw was assessed on a wire mesh: 0 to 3 (0 = normal grip strength; 1 = mildly reduced grip strength; 2 = moderately reduced grip strength; 3 = severely reduced grip strength). Total score was calculated by combining scores of swelling and grip strength of all four paws. Inclusion criteria for the hTNF[Tg/+] control group was a combined clinical score of over 5, at 10 weeks of age.

**Histological analyses.** Hind paws were fixed in 4.5% formalin for 6 h and then decalcified in 14% pH 7.2 EDTA/ammonium hydroxide buffer (Sigma #318604) at 4 °C until bones were pliable. Afterwards, 2 μm decalcified paraffin-embedded sections were prepared and stained with haematoxylin and eosin (H&E) and TRAP (#387A, Leukocyte Phosphatase Staining Kit, Sigma Diagnostics). In brief, slides were stained for 10 min with 1:5 diluted Meyer's hemalum (Merck #1.09249.0500), rinsed with distilled water, differentiated in 1% HCl ethanol and rinsed again for 10 min. Afterwards, slides were stained in eosin working solution (300 ml 1% Eosin Sigma #318906, 600 ml distilled water, 0.1 ml acetic acid 100%) for 15 min. Slides were then rinsed with distilled water, 96% ethanol, *n*-Butyl-acetate and mounted. Regarding TRAP staining, slides were stained for 1 h at 37 °C with the TRAP staining solution (250 μl naphthol AS-BI phosphoric acid, 1000 μl acetate solution, 500 μl tartrate solution, 45 ml distilled water; Sigma-Aldrich #387A) protected from light. Afterwards, slides were developed for 2 min at 37 °C with a mix of 250 μl of Fast garnet GBC base solution and 250 μl of sodium nitrite solution. Nuclei were stained with Meyer's hemalum as described above and slides mounted in Aquatex (Merck #108562). Sections were analysed using an Axioskop 2 microscope (Carl Zeiss MicroImaging) and Osteomeasure Analysis System (OsteoMetrics) to quantify the areas of inflammation, erosion and osteoclast numbers.

**Ctsk in vivo imaging.** Cat K 680 FAST (Perkin Elmer #NEV11000) was reconstituted and 100 μl per animal injected intravenously according to the manufacturer's guidelines. Ctsk intensity (Avg Radiant Efficiency) was determined by IVIS® Lumina Series III (Perkin Elmer) 24 h post injection.

**In vitro restimulation of spleen cells with collagen.** Spleen cells were harvested, passed through a nylon mesh, and cultured at a density of $2 × 10^6$ cells/ml in RPMI 1640 (L-glutamine, 10% FCS, penicillin/streptomycin, β2-mercaptoethanol) and stimulated with 100 μg/ml of chicken type II collagen (Sigma-Aldrich #C9301) for 72 h. During the last 18 h of culture, cells were incubated with 1 μCi/well of ³H-thymidine, to quantify proliferation using a scintillation counter (Beckman).

**Osteoclastogenesis and staining.** Wildtype animals (C57BL/6J) used as bone marrow donors were bred at the Medical University Vienna. Hematopoietic stem cells of the bone marrow were isolated and cultured in complete MEMα (Gibco #32561037) containing 5% Pen-Strep (Gibco #15140122) and 10% foetal calf serum (FCS, Gibco #10082147) supplemented with 100 ng/ml M-CSF (R&D Systems #416). After 3 days, cells were harvested, plated and cultured in Full-MEMα supplemented with 30 ng/ml M-CSF and/or 50 ng/ml RANKL (R&D Systems #462) for another 3–4 days including one medium change on day 6. Osteoclasts were defined as multinucleated cells (≥3 nuclei) with TRAP positivity (Sigma-Aldrich #387A). RecArg1 was used in vitro at 1 μg/ml starting at day 3 of osteoclastogenesis, unless otherwise stated. Arg-Free and AA-free media (both Gibco custom MEMα based on 32561037) was supplemented with dialyzed FCS (Gibco #26400036) and AAs and used starting day 3 of osteoclastogenesis. Preosteoclasts in tryptophan and cysteine-free medium were cultured in 5% dialyzed FCS to control for serum-bound AAs[44] (Supplementary Table 1). 2-(difluoromethyl) ornithine (DFMO) was a kind gift from the Woster Lab (MUSC) and used at a concentration of 2.5 mM. *Tsc2*[fl/fl] and *Rictor*[fl/fl] LysMcre[+/−] have previously been

described[45,46]. *Eif2ak4*[−/−] (B6.129S6-*Eif2ak4*[tm1.2Dron]/J) and *Arg1*[fl/fl]Tie2[+/−] bone marrow was provided by the Murray Lab[17,44]. *Ddit3*[−/−] (B6.129S(Cg)-*Ddit3*[tm2.1Dron]/J) animals were acquired from the Jackson Laboratories. *Slc38a9*[−/−] (unpublished) bone marrow was a kind gift from the Surperti-Furga Lab[20]. Human monocytes were isolated from whole blood using Histopaque-1077 and Histopaque-1119 (Sigma #10771 and #11191) and CD14+ cells sorted using CD14-PerCP-Cy5.5 (eBioscience #45-0149, 61D3). Monocytes were resuspended in complete MEMα medium (5% Pen-Strep, 10% FCS) with 25 ng/ml M-CSF. After overnight incubation with M-CSF, human RANKL (R&D Systems #390) was added at 25 ng/ml for another 3–4 days including one medium change on day 6. Osteoclasts were defined as multinucleated cells (≥3 nuclei) with TRAP positivity. Blood from healthy donors included in this study was provided by the Biobank for patients with rheumatic diseases and were approved by the Clinical Research Ethics Committees of the Medical University Vienna (EK Nr. 559/2005). An informed consent was obtained from all subjects.

**Giant cell formation and staining.** Bone marrow cells were cultured for 3 days in Full-MEMα containing 15% v/v L929-supernatant on 6-well non-treated plastic plates. Adherent macrophages were detached using 1× PBS (10 mM EDTA). Cells were plated in Permanox® 8-well chamber slides in Full-MEMα containing 15 ng/ml IL-4 (R&D Systems #404) and described stimuli for another 3 days, after which the medium was renewed. Three days later, cells were stained using H&E. H&E-stained slides were analysed using an Axioskop 2 microscope (Carl Zeiss MicroImaging). For nuclei counts, cells were washed with PBS and stained (0.5% PFA, 0.1% Triton X-100, 0.1% Hoechst 33342, 0.15% Phalloidin AlexaFluor488 in 1× PBS) for 20 min in the dark. Afterwards, cells were washed again and kept in PBS for imaging. Twenty-five random images per well were taken and analysed using Cell Profiler[47].

**Macrophage and dendritic cell differentiation.** Macrophage or dendritic cells were differentiated from hematopoietic stem cells cultured in presence of 30 ng/ml M-CSF (macrophages) or 20 ng/ml granulocyte macrophage colony-stimulating factor (GM-CSF, R&D #215 DCs) and 5 ng/ml IL-4 (dendritic cells), with complete medium changes on days 3 and 6. On day 7, cells were harvested and analysed using flow cytometry.

**Quantitative PCR.** Total RNA was extracted from paws and cells using RNA isolation kits (QIAGEN RNeasy; Peqlab Trifast). Reverse-transcription was performed using commercially available kits (Applied Biosystems). SYBR Green Supermix (Bio-Rad Laboratories) was used for the qPCR reaction. Postamplification melting curve analysis and water controls were included to ensure absence of primer dimers. To obtain sample-specific ΔCt values, normalization to hypoxanthine phosphoribosyltransferase 1 (*Hprt*) within each sample was performed. Data are shown as fold change, where $2^{−ΔΔCt}$ values were calculated (ΔΔCt = ΔCt treatment − ΔCt control). RT-PCR was performed using the following primers: *Hprt*: 5′-CGCAGTCCCAGCGTCGTG-3′ and 5′-CCATCTCCTTCATGACATC TCGAG-3′; *Nfatc1*: 5′-GACAGACATCGGGAGGAAGA-3′ and 5′-AGCCTTCT CCACGAAAATGA-3′; *Ctsk*: 5′-GGAAGAAGACTCACCAGAAGC-3′ and 5′-GT CATATAGCCGGCCTCCACAG-3′; *Acp5*: 5′-ACAGCCCCCACTCCCACCCT-3′ and 5′-TCAGGGTCTGGGTCTCCTTGG-3′; *Tnf*: 5′-CCACCACGCTCTTCTG TCTAC-3′ and 5′-AGGGTCTGGGCCATAGAACT-3′; *Fos*: 5′-AGCCCAGACCT GCAGTGGCT-3′ and 5′-GCGCTCTGCCTCCTGACACG-3′; *Mt-ND2*: 5′-AGGG ATCCCACTGCACATAG-3′ and 5′-5CTCCTCATGCCCCTATGAAA-3′; *Arg1*: 5′-GGAAAGCCAATGAAGAGCTG-3′ and 5′-GCTTCCAACTGCCAGACTGT-3′; *Dcstamp*: 5′-TCCTCCATGAACAAACAGTTCCAA-3′ and 5′-AGACGTGGTT TAGGAATGCAGCTC-3′; *Sdhd*: 5′-TGGTCAGACCCGCTTATGTG-3′ and 5′-GGTCCAGTGGAGAGATGCAG-3′; *Pck2*: 5′-TGCCAGGCTGGAAAGTGGAG TGT-3′ and 5′-GCAACCCCAAAGAAGCCGTTCTCA-3′.

**Western blot.** Cells were washed with ice-cold PBS, scraped and centrifuged at $400 × g$ for 5 min at 4 °C and the resulting pellet was lysed with lysis buffer ((20 mM Hepes pH 7.4, 400 mM NaCl, 25% v/v glycerol, 1 mM EDTA, 0.5 mM NaF, 0.5 mM Na₃VO₄, 0.5 mM DTT) supplemented with Triton-X, PMSF, PIM and RPI shortly before use). The homogenate was cleared by centrifugation at 4 °C for 10 min at $16,000 × g$ and the supernatant containing the protein fraction recovered. Protein concentration in the supernatant was determined using the Pierce BCA Protein Assay Kit (Thermo Fisher Scientific #23225). A total of 15 μg of proteins were resolved by SDS-PAGE and transferred to PVDF membranes (GE Healthcare #10600023). Membranes were blocked with 1% Western Blocking Reagent (Roche #11096176001) and incubated with primary antibodies at 4 °C overnight. The following antibodies were used: p-pS6K T389 (Cell Signaling #9234, 108D2, 1:1000); 4EBP1 (Cell Signaling #9452, 1:500); p-mTOR Ser2448 (Cell Signaling #5536, D9C2, 1:1000); total S6K (Cell Signaling #2708, 49D7, 1:1000); Grb2 (BD #610112, 81, 1:1000), p-eIF2a XP Ser51 (Cell Signaling #3398, D9G8, 1:1000), p-4EBP1 Thr37/46 (Cell Signaling #2855, 236B4, 1:500), Tubulin (Cell Signaling #3873, DM1A, 1:1000), ARG1 (Merck #ABS535, 1:000) and Actin (Sigma Aldrich #A2066, 1:1000). Incubated membranes were washed three times for 5 min with 0.5% Western Blocking Reagent and probed with the appropriate anti-igG-horseradish peroxidase-linked (HRP) secondary antibody (GE Healthcare #NA934,

anti-rabbit IgG, 1:20,000; #NA931, anti-mouse IgG, 1:20,000; Promega #G135A, anti-chicken IgY, 1:20,000). Antigen-specific binding of antibodies was detected with SuperSignal West Femto and Pico Kits (Thermo Scientific #34095 and #34577).

**Metabolism assays**. OCR and extracellular acidification rate (ECAR) measurements were performed on a Seahorse XFe96 Analyzer (Agilent) using the Seahorse XF Cell Mito Stress test kit (Agilent #103015-100) according to the manufacturer's instructions. In brief, 150,000 preosteoclasts were seeded per well and treated with respective stimuli. During the measurements, oligomycin (1 μM), carbonyl cyanide-p-trifluoromethoxyphenylhydrazone (FCCP, 1 μM) and rotenone/antimycin A (500 nM) were subsequently injected. Raw data were analysed using Wave Desktop Software (Agilent, version 2.6.1) and exported and graphed in GraphPad Prism 8 (GraphPad Software).

**Flow cytometry**. Plated preosteoclasts were harvested in Accutase solution (Sigma Aldrich #A6964) and later stained for DAPI (Biolegend #422801, 1:200), fixable viability dye (Invitrogen #65-0865-14, 1:2000), MitoTracker Green (Thermo Fisher #M7514, 100 nM) and F4/80-BV421 (BioLegend #123131, BM8, 1:80) according to the manufacturer's instructions. Bone marrow dendritic cells and macrophages were FC blocked with TruStain fcX (BioLegend #101320, 1:200) and surface stained for CD45.2-APC (TONBO #20-0454, 104, 1:80), CD11c-APC-eF780 (eBioscience #47-0114, N418, 1:50), CD11b-PeCy7 (eBioscience #25-0112, Clone M1/70, 1:400), MHC Class II-PE (TONBO #50-5321, M5/114.15.2, 1:80) and F4/80-BV421. For in vivo assessment of osteoclast precursors, mashed splenocytes were stained for CD45.2-APC, F4/80-FITC (BioLegend #123108, BM8, 1:80), CD11b-PeCy7 and GR-1-PE (Ly-6G/Ly-6C, BioLegend #108408, RB6-8C5, 1:80). Cells were acquired and analysed using CytoFLEX S Flow Cytometry (Beckman Coulter), CytExpert (Version 2.0) and FlowJo (Version 10, LLC) software. MFI represents mean fluorescence intensity for Fig. 4e.

**Bioinformatics analysis**. All bioinformatics analyses and visualizations have been conducted using R version 3.5.1, ggplot2 3.1.0 and heatmaply 0.15.2 (ref. [48]). Network analysis relied on OmicsIntegrator 0.3.1 (ref. [49]) version 1.0 of the python implementation of Infomap[50] and Cytoscape 3.6.1 (ref. [51]). All the code and software parameters used for this study are provided through Jupyter notebooks on a Github repository.

**Transcriptomics**. Total RNA was prepared from $1 \times 10^6$ preosteoclasts using TRIzol Reagent (Thermo Fisher Scientific, #15596026). A total of 200 ng of total RNA was subsequently utilized for RNA-Seq library preparation by using TruSeq SR RNA sample prep kit (Illumina, #FC-122–1001), following the manufacturer's protocol. The libraries were sequenced for 50 cycles (single read) with a HiSeq 2000 (Illumina). Raw sequencing data were processed with CASAVA 1.8.2 to generate FastQ files. Sequence reads were mapped onto the mouse genome build mm9 using TopHat 2.0 (ref. [52]). From the RNA-Seq alignments files, read counts have been obtained for mm9 using Rsubread's featureCount[53], with and without allowing reads mapping multiple genes and counting overlapping reads for multiple genes. The results were similar using both methods. For instance, the Jaccard coefficient was equal to 0.837 when looking at genes differentially expressed between RANKL and RANKL/Arg-Depletion conditions with and without allowing for multimapping reads. Therefore, of the 469 genes found to be differentially expressed with or without read multimapping, 393 were common. The differential expression analysis was thus first based solely on counts obtained allowing multimapping reads (Figs. 2b, 3b, c, 4g and Supplementary Fig. 2). Genes were filtered to keep only those with log(CPM + 0.25) > 0 in at least three samples. Library sizes were scaled, values were transformed and variance corrected in order to fit a linear model explaining the read abundances using the voom method[54] before identifying differentially expressed genes between conditions, with a log2 fold-change higher than 1 for a false discovery rate (FDR) of 5% using limma[55] and edgeR[56]. Hierarchical clustering with complete linkage of the z-score-transformed expression values for the 40 genes with highest difference between RANKL and RANKL/Arg-Depletion segregated all samples by experimental conditions. The gene sets identified were tested for enrichment using the clusterProfiler library[57] for the different levels of the Gene Ontology (GO)[58,59] and KEGG[60,61] in *Mus musculus*. Moreover, genes were filtered on membership to metabolism (mmu:09100) or KEGG pathways of interest (mmu00010, mmu00020, mmu00220, mmu00330) using the REST API provided. Pairwise comparisons in expression levels between conditions were assessed using a permutation test, with the null hypothesis that there were no associations between the expression of any gene of the pathway and the conditions compared. Samples were ranked on their expression level for each gene in the pathway and the absolute difference between the sum of the ranks of the four replicates in each condition were summed over all genes, defining the test statistic. The association between samples and conditions was randomized 12,000 times and the same value computed, resulting in an empirical P-value by comparing the observed statistic to the simulated distribution. These values were reported after Bonferroni correction. The intersection of genes differentially expressed both with and without counting multimapping reads was then used to generate robust gene sets for the multi-omics integration analysis.

**Proteomics**. Cell pellets were lysed in 8 M urea, 10 mM HEPES (pH 8), 10 mM DTT and sonicated at 4 °C for 15 min (level 5, Bioruptor, Diagenode). Alkylation of reduced cysteines was performed in the dark for 30 min with 55 mM iodacetamide (IAA) followed by a two-step proteolytic digestion. Samples were digested at 21–24 °C with LysC (1:50, w/w, Wako) for 3 h. Cell lysates were adjusted to 2 M Urea with 50 mM ammoniumbicarbonate and then both cell lysates and supernatants were digested with trypsin (1:50, w/w, Promega) at 21–24 °C overnight. The resulting peptide mixtures were acidified and loaded on C18 StageTips (EmporeTM, IVA-Analysentechnik). Peptides were eluted with 80% acetonitrile (ACN), dried using a SpeedVac centrifuge, and resuspended in 2% ACN, 0.1% trifluoroacetic acid (TFA) and 0.5% acetic acid. Chemicals were purchased from Sigma-Aldrich unless stated otherwise. For ultra-high pressure LC–MS, peptides were separated on an EASY-nLC 1200 HPLC system (Thermo Fisher Scientific) coupled online to the Q Exactive HF-X mass spectrometer via a nanoelectrospray source (Thermo Fisher Scientific)[62]. Peptides were loaded in buffer A (0.5% formic acid) on in house packed columns (75-μm inner diameter, 50 cm length and 1.9 μm C18 particles from Dr. Maisch GmbH, Germany). Peptides were eluted with a nonlinear 170-min gradient of 5–60% buffer B (80% ACN, 0.5% formic acid) at a flow rate of 300 nl/min and a column temperature of 55 °C. The Q Exactive HF was operated in a data-dependent mode with a survey scan range of 300–1650 $m/z$ and a resolution of 60,000–120,000 at $m/z$ 200. Up to the ten most abundant isotope patterns with a charge > 1 were isolated with a 1.4 Thomson (Th) isolation window and subjected to higher-energy collisional dissociation (HCD) fragmentation at a normalized collision energy of 27. Fragmentation spectra were acquired with a resolution of 15,000 at $m/z$ 200. Dynamic exclusion of sequenced peptides was set to 30 s to reduce repeated peptide sequencing. Thresholds for ion injection time and ion target values were set to 20 ms and 3E6 for the survey scans and 60 ms and 1E5 for the MS/MS scans, respectively. Data were acquired using the Xcalibur software (Thermo Scientific). MaxQuant software (version 1.5.3.2) was used to analyse MS raw files[63]. MS/MS spectra were searched against the human Uniprot FASTA database (Version July 2015, 91645 entries) and a common contaminants database (247 entries) by the Andromeda search engine[64]. Cysteine carbamidomethylation was applied as fixed and N-terminal acetylation, deamidation at NQ, and methionine oxidation as variable modifications. Enzyme specificity was set to trypsin with a maximum of 2 missed cleavages and a minimum peptide length of 7 AAs. An FDR of 1% was applied at the peptide and protein level. Peptide identification was performed with an allowed initial precursor mass deviation of up to 7 ppm and an allowed fragment mass deviation of 20 ppm. Nonlinear retention time alignment of all measured samples was performed in MaxQuant. Peptide identifications were matched across all samples within a time window of 1 min of the aligned retention times. A library for 'match between runs' in MaxQuant was built from duplicate and additional single shot MS runs from MACS enriched cell types. Protein identification required at least 1 'razor peptide' in MaxQuant. A minimum ratio count of 1 was required for valid quantification events via MaxQuant's Label Free Quantification algorithm (MaxLFQ). Downstream bioinformatic analysis was conducted in the Perseus environment[65]. Each quantified protein was required to be identified in three out of four replicates of at least one condition. Protein LFQ intensities were logarithmised to the base 2 and missing values imputed from a random normal distribution centred on the detection limit. Data were imported into Perseus, filtered to keep only proteins where the coefficient of variation of overall abundance was greater than the coefficients of variation in every condition, to select for proteins that consistently vary between conditions but not between replicates, and converted to z-scores. Average abundances among different conditions were compared by a two-sided Student's t-test (permutation-based FDR = 0.05, s0 = 0.05) and used to select proteins with differential abundance between conditions. As in the transcriptomic analysis, hierarchical clustering with complete linkage of the z-score-transformed data segregated the samples by experimental conditions.

**Network integrative analysis**. OmicsIntegrator was used to integrate transcriptomic and proteomic changes, by addressing the underlying molecular mechanisms of the effects of recArg1 and the arginine-dependent effects of RANKL[49]. This tool circumvents the low overlap between proteomic and transcriptomic hits (Supplementary Fig. 6b), by inferring transcription factors likely to explain the transcriptional changes and searching for minimal sets of connected genes including these transcription factors and abundantly changed proteins. The first set of proteins and genes, describing the effect of recArg1, were defined by significantly changed abundance or expression respectively between RANKL/Arg-Rescue and RANKL/Arg-Depletion for proteomics data, and between RANKL and RANKL/Arg-Depletion for transcriptomics data. The second set, describing the arginine-dependent effect of RANKL, was defined as the exclusive disjunction of proteins or genes with changes in abundance or expression between RANKL-positive and RANKL-negative conditions in the presence and absence of arginine. By doing so, we selected for genes differentially expressed between M-CSF and RANKL or M-CSF/Arg-Rescue and RANKL/Arg-Rescue but not between M-CSF/Arg-Starvation and RANKL/Arg-Starvation, and proteins with different abundance between M-CSF/Arg-Rescue and RANKL/Arg-Rescue but not between M-CSF/Arg-Starvation and RANKL/Arg-Starvation. The protein interactions used were mouse–mouse interactions from BioGRID Release 3.4.160 (ref. [66]), weighted by directness and experimental strength of the assay (Supplementary Table 2). The

garnet tool was used to identify transcription factors explaining transcriptomic variability, converted to up-to-date murine identifiers and integrated with hit proteins via the Forest tool. The overlap between both resulting Steiner trees was contextualized by randomly selecting 2000 connected subgraphs of matched size on the BioGRID network and examining the corresponding shared nodes. Infomap algorithm and its python implementation[50] were used to cluster nodes into modules (two-level clustering) with a scaling factor of the link flows allowing a limited number of communities (--markov-time 2.5), which were visualized with the map generator[67]. Major network structures were identified according to coding theory, by looking at which grouping of the nodes allows the minimal coding of trajectories of random walks on the network (Supplementary Table 3). The information flow inside and between these groups therefore represents how tightly connected nodes are inside and between these groups. The resulting communities were analysed for enrichment in GO Biological processes.

**Metabolomics and $^{13}C_6$ arginine tracing**. Cells were cultured for 24 h in the presence of fully labelled arginine ($^{13}C_6$ arginine, Sigma Aldrich). Medium was collected and 10 μl of $^{13}$C-glycerol (150 μl/ml) was added as internal standard. Cells were scrapped, collected and frozen. Cell pellets were resuspended with 300 μl of cold methanol/water (8:1, v/v) containing $^{13}$C-glycerol (5 μl/ml) as internal standard. Metabolites were extracted with three rounds of liquid $N_2$ immersion and sonication, followed by 1 h in ice before centrifugation at 22,800 × g (10 min at 4 °C). Medium was collected and 10 μl of $^{13}$C-glycerol (150 μl/ml) was added as internal standard. The medium samples were lyophilized and resuspended in 500 μl of cold methanol/water (8:1, v/v). After vortexing, samples were left 1 h in ice and centrifuged 10 min at 22,800 × g (4 °C). Metabolite extraction of cells and medium was split into two aliquots of 250 μl (cells) or 400 μl (medium) for gas chromatography–MS (GC–MS), and 40 μl (both) for LC–MS analysis. For GC–MS analysis, samples were dried under a stream of $N_2$ gas and lyophilized before chemical derivatization with 40 μl of methoxyamine in pyridine (30 μg/ml) for 45 min at 60 °C. Samples were also silylated using 25 μl of N-methyl-N-tri-methylsilyltrifluoroacetamide with 1% trimethylchlorosilane (Thermo Fisher Scientific) for 30 min at 60 °C to increase volatility of metabolites. A 7890A GC system coupled to a 7000 QqQ mass spectrometer (Agilent Technologies) was used for isotopologue determination. Derivatized samples were injected (1 μl) in the gas chromatograph system with a split inlet equipped with a J&W Scientific HP-5ms stationary phase column (30 m × 0.25 mm i.d., 0.1 μm film, Agilent Technologies). Helium was used as a carrier gas. The temperature gradient was the following: from 70 to 150 °C at a heating rate of 5 °C/min, from 150 to 250 °C at 10 °C/min and from 250 to 325 °C at 50 °C/min. Metabolites were ionized using positive chemical ionization (CI) with isobutene as reagent gas. Mass spectral data on the 7000 QqQ were acquired in scan mode monitoring selected ion clusters of the different metabolites (Supplementary Table 4). For LC–MS analysis, samples were analysed using an UHPLC system coupled to a 6490 QqQ mass spectrometer (Agilent Technologies) to determine arginine. Cells and medium extracts were injected (5 μl and 1 μl, respectively) and metabolites separated using an ACQUITY UPLC HSS T3 column (1.8 μm, 2.1 × 150 mm, Waters). The mobile phases used for the separation of the metabolites were A: water with 0.1% formic acid and B: ACN with 0.1% formic acid. The chromatographic gradient was isocratic for 2 min at 100% A, and from minute 2 to 3 decreased to 10% A. From minute 3 to 4, the percentage of A raised again to 100% and finally the column was equilibrated at 100% A until 8 min. Flow rate was 0.3 ml/min. The QqQ mass spectrometer worked in MRM mode using the transitions 181 → 74 (CE:20V) and 181 → 121 (CE:8V) to determine labelled Arg. The electrospray ionization source (ESI) worked in positive mode.

**$^{13}C_6$ arginine and $^{13}C_4$ aspartate tracings**. Cells were cultured for 24 h in the presence of fully labelled arginine ($^{13}C_6$ L-arginine, Cambridge Isotope Laboratories #CLM-2265) and fully labelled aspartate ($^{13}C_4$ L-aspartic acid, Cambridge Isotope Laboratories #CLM-1801). Cell extracts where where centrifuged for 10 min at 5000 × g. The supernatant was collected and dried using nitrogen evaporator. The samples were reconstituted in 50 μl of methanol, centrifuged for 10 min at 1000 × g and supernatant was used for LC–MS analysis. For AA tracing, a Vanquish UHPLC system (Thermo Scientific) coupled to an Orbitrap Fusion Lumos (Thermo Scientific) mass spectrometer was used for the LC–MS analysis. The chromatographic separation for samples was carried out on an ACQUITY UPLC BEH Amide, 1.7 μm, 2.1 × 100 mm analytical column (Waters) equipped with a VanGuard: BEH Amide, 2.1 × 5 mm pre-column (Waters). The column was maintained at a temperature of 40 °C and 2 μl sample were injected per run. The mobile phase A was 0.15% formic acid (v/v) in water and mobile phase B was 0.15% formic acid (v/v) in 85% ACN (v/v) with 10 mM ammonium formate. The gradient elution with a flow rate 0.4 ml/min was performed with a total analysis time of 17 min. The mass spectrometer was operated in a positive electrospray ionization mode: spray voltage 3.5 kV; sheath gas flow rate 60 arb; auxiliary gas flow rate 20 arb; capillary temperature 285 °C. For the analysis a full MS scan mode with a scan range m/z 50–250, resolution 500,000; AGC target 2e5 and a maximum injection time 50 ms was applied. The data processing was performed with the TraceFinder 4.1 software (Thermo Scientific). For TCA cycle tracing, a Vanquish UHPLC system (Thermo Scientific) coupled to an Orbitrap Fusion Lumos (Thermo Scientific) mass spectrometer was used for the LC–MS analysis. The chromatographic separation for samples was carried out on an ACQUITY HSS T3, 1.8 μm, 2.1 × 100 mm analytical

column (Waters) equipped with a VanGuard HSS T3, 2.1 × 5 mm pre-column (Waters). The column was maintained at a temperature of 40 °C and 2 μl of sample was injected per run. The mobile phase A was 0.1% formic acid (v/v) in water and mobile phase B was 0.1% formic acid (v/v) in methanol. The gradient elution with a flow rate 0.5 ml/min was performed with a total analysis time of 10 min. The mass spectrometer was operated both in positive and negative ionization mode: spray voltage for positive mode 3.5 kV and 3.0 kV for negative mode; sheath gas flow rate 60 arb; auxillary gas flow rate 20 arb; capillary temperature 285 °C. For the analysis a full MS scan mode with a scan range m/z 80–400, resolution 500,000; AGC target 2e5 and a maximum injection time 50 ms was applied. The data processing was performed with the TraceFinder 4.1 software (Thermo Scientific).

**Serum metabolomics**. Serum concentrations of arginine were assessed using an arginine ELISA Kit (#K 207733, IDK). For serum samples, 10 μl of sample was placed on a 96-well hydrophobic filter plate and mixed with 10 μl of an isotopically labelled internal standard mixture. A total of 300 μl of methanol was added and the plate was shaken for 20 min at 450 rpm. Afterwards, the sample extract was collected in a 96-well plate by centrifuging the filter plate for 5 min at 500 × g. The sample extracts were used for LC–MS analysis. A Vanquish UHPLC system (Thermo Scientific) coupled with an Orbitrap Q Exactive (Thermo Scientific) mass spectrometer was used for the LC–MS analysis. The chromatographic separation for samples was carried out on an ACQUITY UPLC BEH Amide, 1.7 μm, 2.1 × 100 mm analytical column (Waters) equipped with a VanGuard: BEH C18, 2.1 × 5 mm pre-column (Waters). The column was maintained at a temperature of 40 °C and the sample injection volume was 2 μl. The mobile phase A was 0.15% formic acid (v/v) in water and mobile phase B was 0.15% formic acid (v/v) in 85% ACN (v/v) with 10 mM ammonium formate. The gradient elution with a flow rate 0.4 ml/min was performed with a total analysis time of 17 min. The Orbitrap Q Exactive (Thermo Scientific) mass spectrometer was operated in an electrospray ionization positive mode, spray voltage 3.5 kV, aux gas heater temperature 400 °C, capillary temperature 350 °C, aux gas flow rate 12. The metabolites of interest were analysed using a full MS scan mode, scan range m/z 50–400, resolution 35,000, AGC target 1e6, maximum IT 50 ms. The Trace Finder 4.1 software (Thermo Scientific) was used for the data processing. Seven-point linear calibration curves with internal standardization and 1/x weighing were constructed for the quantification of metabolites.

**Image processing**. Images were cropped and processed using Adobe Photoshop CS6, adjusting brightness and contrast. Original uncropped Western Blot images are provided in the source data file.

**Statistics**. Statistical analysis was performed using a two-tailed t-test for two groups, an ordinary one-way ANOVA followed by Bonferroni's multiple comparisons test or Tukey post-hoc test for multiple groups and a two-way ANOVA followed by Bonferroni's multiple comparisons test for curve analysis, unless otherwise stated. Statistical outliers for patient data in Fig. 1 have been excluded based on alpha = 0.05 on Prism 8 software (GraphPad, La Jolla, CA). In vitro data are representative of at least two repeats, while transcriptomics, proteomics and metabolomics correspond to biological replicates. Statistical significance is indicated by $*P < 0.05$, $**P < 0.01$, $***P < 0.001$, $****P < 0.001$. All error bars indicate ±SEM.

**Reporting summary**. Further information on research design is available in the Nature Research Reporting Summary linked to this article.

## Data availability

RNA-Seq data have been deposited in Gene Expression Omnibus (GEO) under accession number GSE125101. The mass spectrometry proteomics data have been deposited to the ProteomeXchange Consortium via the PRIDE[68] partner repository with the dataset identifier PXD012405. The source data underlying Figs. 1–2 and 4–7 and Supplementary Figs. 1–4 and 7 are provided as a Source Data file or available from the corresponding authors upon reasonable request.

## Code availability

All code used in the proteomics and transcriptomics differential analyses and subsequent integration is available on GitHub (https://doi.org/10.5281/zenodo.2541644).

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

## Acknowledgements

We thank Tetyana Shvets, Hannah Paar, Lucia Quemada Garrido, Mario Kuttke, Maximilian Kugler and Manuel Salzmann for technical assistance. We thank Giulio Surperti-Furga for the *Slc38a9*$^{-/-}$ mice and George Kollias for providing the hTNF$^{Tg/+}$ mice. This research was funded by the Austrian Society for Rheumatology (ÖGR) (J.S.B.), the FWF (G.S.: 30026, 31106; O.S.: 31568) and the Christian Doppler Laboratory for Arginine Metabolism in Rheumatoid Arthritis and Multiple Sclerosis (G.S.). J.S.B., A.V. and A.L. were rewarded a DOC fellowship by the Austrian Academy of Sciences. A.B. received funding from the European Research Council (ERC) under the European Union's Seventh Framework Program and Horizon 2020 research and innovation program (grant agreement No. 677006, CMIL). M.R., J.B.B. and P.J.M. received funding from the Max Planck Gesellschaft.

## Author contributions

J.S.B., P.J.M., O.S., S.B. and G.S. conceived and designed the study. J.S.B., M.H., M.Ki., A.L., A.V., M.Ke., V.S., B.N., A.J., A.F., C.S., Y.M., K.K., M.R., J.B.B. and O.Y. performed the experiments. L.V. analysed transcriptomic and proteomic data and performed integrated network analysis. Y.M. performed RNA sequencing. A.F. performed proteomics. K.K., O.Y. and A.J. performed metabolomic analysis. J.S.B., L.V., A.L. and A.J. analysed the data. G.K., A.B., J.J.O., T.W., F.M., J.S.S., P.C., J.M., O.Y. and P.J.M. provided key resources. J.S.B., L.V., O.S., S.B. and G.S. wrote the manuscript. All authors read, revised and approved the final manuscript.

## Competing interests

The authors declare the following competing interests: P.C. is the founder of Biocancer Treatment International Ltd. P.C., G.S. and S.B. are listed as inventors on a patent (US9789169B2) covering recArg1/BCT-100. The remaining authors declare no competing interests.
