## [Peer Review File · Nature Communications]

Reviewers' Comments:

Reviewer #1:

Remarks to the Author:

In this manuscript, the authors investigated how adaptive metabolic responses control polykaryon developmental programs. The authors proposed that extracellular arginine is necessary for RANKL induced metabolism in osteoclast differentiation.

This manuscript is potentially interesting because this is the first report showing that extracellular arginine presence controls giant cell metabolism. However, the present study has the critical concerns. Importantly, the authors did not use osteoclast-specific arginase deficient mice. In addition, although the clinical data of recArg1 is now available, the authors did not provide the human data.

Taken together, this manuscript is still immature for publication in Nature Communications in its current form.

Major concerns

1. The importance of arginine in osteoclast differentiation cannot be proven without conditional KO mice. The authors should show the phenotypes of osteoclast-specific arginase deficient mice.
2. The authors should provide more detailed phenotypes that are typically analyzed in the field of bone biology in arginine deletion model (e.g. body size, body weight, serum calcium and phosphorus levels, whole-body X-ray picture, bone morphometric analysis and CT analysis)
3. Because the clinical data of recArg1 is now available, the authors should provide the human data.
4. The authors should check the serum levels of OPG and RANKL of the mice in this study.

Reviewer #2:

Remarks to the Author:

The manuscript by Brunner and colleagues examines the effect of arginine depletion on osteoclast polykaryon formation by a combination of in vitro (studies comprising transcriptomics, proteomics, metabolomics and OXPFOs) and in vivo models of arthritis. The results show that the use of FDA-approved BCT-100 inhibits polykaryon formation and reduces the severity of arthritis. The mechanisms through which this effect occurs are independent of mTOR activation, through a general blunting of RANKL signaling. The same conclusions are drawn for the IL-4 derived giant cells, even though only polykaryon formation and OXPFOs were tested in these cells. The general conclusion stresses the importance of nutrient availability for polykaryon formation, an energy demanding process that controls bone remodelling. The manuscript needs strengthening on two main grounds; the central hypothesis (1) and conclusions reached upon incomplete data (2 and 3).

1. The rationale behind the choice of arginine depletion to study polykaryon formation and arthritis is not clear. Figure 4 shows that polykaryon formation is equally depending on leucine, isoleucine and phenylalanine and Extended Figure 7 shows that osteoclastogenesis does not strictly depend on essential aa (example of Gln). Importantly, metabolism of 13/20 aa controls osteoclastogenesis. For those 13 where commercial inhibitors like BCT-100 are available, the effect on the mice CIA should be explored. This is particularly important since there is no mechanism (signalling, transcriptional, metabolic or proteomic) which could be specifically attributed to arginine depletion in RANKL-dependent osteoclast formation.

2. The metabolomics experiments are incomplete and a TCA rewiring cannot be concluded based on the presented data. ¹³C arginine is used to measure cellular uptake (Figure 3g) but this does not inform on the specific contribution of labelled arginine to the TCA cycle. Targeted metabolomics with labelled arginine would allow an accurate contribution of this aa into TCA metabolites in osteoclasts. These experiments should be carried out in presence and absence of BCT-100. The TCA rewiring is concluded based on gene expression data in serine/purine synthesis pathways. Isotope tracing experiments will confirm if this is the case. Furthermore, the Seahorse experiments are performed 48h after RANKL stimulation, whereas metabolomics after 24h. This is problematic because the manuscript makes parallels between OXPFOs effect and metabolomics results, despite the fact that osteoclasts could be at different stages of their differentiation.

3. The IL-4 derived giant cells data are incomplete if one would like to conclude similarities with osteoclasts in the abstract and title. Tracing experiments with labelled arginine will be necessary to claim that a similar metabolic adaptation occurs. The manuscript claims that arginine depletion specifically abolishes RANKL signaling in osteoclasts. If same is true in IL-4 signaling, then this treatment has pleiotropic effects and is not pathway-specific as suggested.

Minor

1. There are many shortcuts in reaching conclusions. To discriminate between inflammation and osteoclast driven damage in arthritis models, requires genetic approaches (Ctsk-driven conditional KO). Mentioning an osteoclast-specific effect of the drug is an over-statement and should be avoided.

2. Line 186-206. The paragraph has speculative statements. There is no data to support aa transient pre-osteoclast state.

3. The text is difficult to follow at times (e.g. page 8, line 164; the sentence starting with 'subtle effects' needs revising)

4. Alpha-ketoglutarate is misspelled on numerous occasions, including in figures (Figure 3)

5. Line 238-241. The general conclusion is highly speculative and does not reflect the data presented in the manuscript.

6. Given the major differences in urea cycle between humans and mice (absence of nitric oxide synthase induction in inflammatory human mononuclear phagocytes), it is not accurate to claim in the title that 'metabolic adaptation controls multinucleated giant cell formation'. The manuscript shows data on the metabolism of murine osteoclast formation and it is unclear if the same applies to the IL-4-induced polykaryons and to human osteoclasts and giant cells.

Reviewer #3:

Remarks to the Author:

Brunner et al demonstrate that the administration of recombinant Arginase-1 (recArg1) inhibits osteoclastogenesis and improves disease outcome in several arthritis mouse models. They further show that reducing systemic arginine levels does not globally affect the immune system but quite specifically affects the formation of osteoclasts.

In a series of in vitro experiments, the authors demonstrate that RANKL-mediated osteoclast differentiation was associated with a strong increase in the arginine biosynthesis pathway. Consistent with an important role for arginine metabolism in osteoclast differentiation, the addition of recArg1 to the culture medium inhibited the differentiation of osteoclasts, whereas recArg1 only exerted minor effects on macrophage and DC differentiation.

Based on a detailed transcriptomics analysis, the authors inferred that recArg1 inhibited osteoclast differentiation through depletion of arginine rather than through the increased generation of downstream metabolites such as urea and ornithine. The authors further show that arginine depletion by recArg1 inhibits RANKL-induced effects on preosteoclasts but does not globally reduce translation, which one would perhaps expect given that arginine is required for protein synthesis.

In the second part of the manuscript, Brunner et al report that the absence of arginine hampered osteoclast differentiation independent of mTOR signaling components or putative arginine sensors. Through a series of experiments addressing the cellular metabolism of preosteoclasts the authors conclude that arginine catabolism sustains OXPHOS. In addition, the authors show that precursors of arginine or downstream metabolites of other essential amino acids can compensate for the absence of the respective amino acid.

Overall, this is an interesting study and the finding that administration of recArg1 attenuates arthritis has translational potential. The figures are clear and well presented. However, the manuscript is sometimes difficult to understand probably because the format is rather short. Some of the experiments that were performed to study metabolism in preosteoclasts need further clarification (detailed below).

Specific comments:

1.) Some parts of the manuscript and figures require a more detailed description:

- For example, the authors state: "RANKL-induced bursts in preosteoclast SRC were counteracted by recArg1". From this one would conclude that the third condition in Fig 3c is "RANKL + recArg1" but it is just labeled with recArg1. It would need clarification if only recArg1 was added. All panels throughout figure 3 were labeled this way.

- There are several sentences that are rather short and difficult to understand. For example, the authors state "RANKL induced glycolysis irrespective of glucose uptake and arginine presence" (Line 161). Only after studying the figure it became clear to me that RANKL enhances the glycolytic rate in preosteoclasts without markedly increasing glucose uptake...In this reviewer's opinion, several sentences should be rewritten to help the reader easily understand the results. The figures, however, are very nice and easy to understand.

- Another example is the following sentence:

"Subtle effects were observed on other AAs (e.g. glycine, alanine), suggesting specific arginine requirements for RANKL signalling, corroborated by enhanced ¹³C labelled arginine intracellular build-up and extracellular arginine uptake from the supernatant, disassociated from arginine transporter transcription (Slc7a1, Slc7a2)"

How does enhanced arginine uptake corroborate the previous observation?

- The authors state "RecArg1 caused intracellular arginine build-up...(Figure 3f, h)". Looking at Fig 3f. it seems that both conditions labeled as "RANKL" and "RecArg1" increase intracellular arginine levels to the same extent. Why is only RecArg1 mentioned and is this condition RANKL + recArg1 or only recArg1?

2.) In Figure 3f, perhaps the most striking differences between the two conditions "RANKL" and "recArg1" were in putrescine and spermidine pointing to polyamine metabolism. Given that the authors report that recArg1 inhibits osteoclast differentiation, it might be interesting to address whether a decrease in polyamines - which are known to affect proliferation and differentiation - plays a role in osteoclastogenesis.

3.) Based on transcriptomics analyses, the authors inferred that recArg1 inhibited osteoclast

differentiation through depletion of arginine rather than through the increased generation of downstream metabolites such as urea and ornithine. This could be easily validated by adding urea or ornithine to the culture medium.

4.) The authors mention that the urea cycle and the TCA are linked through fumarate and conclude that arginine catabolism sustains RANKL-dependent OXPHOS. This might leave the reader somehow with the impression that metabolites of the urea cycle are shunted into the TCA cycle to enhance OXPHOS but there is no evidence for this and would require metabolic tracing (flux) experiments. In this reviewer's opinion, this should be either experimentally addressed or discussed in a separate section.

5.) The idea that catabolic products directly fuel the TCA cycle is taken up again in the last section where the authors conclude: "Importantly, these data suggest a common theme by which osteoclasts dynamically metabolically adapt to the absence of environmental AAs by utilizing their respective TCA fuelling intermediates". This conclusion is based on the finding that arginosuccinate or citrulline can compensate for the absence of arginine. However, this is expected as those are the precursors of arginine and can be readily converted into arginine by the cells. As such, this does not provide evidence that citrulline and arginosuccinate sustain the TCA cycle.

The author's conclusion that amino acid derivatives fuel the TCA cycle is further based on the observation that alpha-ketoisocaproate can compensate for the absence of leucine. The conversion of leucine into alpha-ketoisocaproate is catalyzed by BCAT1/2, yet this is a reversible reaction. Hence, if alpha-ketoisocaproate is added to the culture medium it is readily converted to leucine by the cells. If amino acids were only required to fuel the TCA cycle, then the addition of TCA cycle metabolites, such as a-KG should compensate for the absence of essential amino acids.

The described experiments recapitulate known biochemistry but do not provide enough evidence to conclude that "scarcity in any single essential AAs can be compensated by its derivatives or precursors to sustain oxidative metabolism". (line 239)

Reviewer #1 (Remarks to the Author):

In this manuscript, the authors investigated how adaptive metabolic responses control polykaryon developmental programs. The authors proposed that extracellular arginine is necessary for RANKL induced metabolism in osteoclast differentiation. This manuscript is potentially interesting because this is the first report showing that extracellular arginine presence controls giant cell metabolism. However, the present study has the critical concerns. Importantly, the authors did not use osteoclast-specific arginase deficient mice. In addition, although the clinical data of recArg1 is now available, the authors did not provide the human data.

Taken together, this manuscript is still immature for publication in Nature Communications in its current form.

Response:

We thank Reviewer #1 for his/her interest and for highlighting the novelty of our study. We have addressed the critical concerns of the reviewer in our detailed response below.

Major concerns

1. The importance of arginine in osteoclast differentiation cannot be proven without conditional KO mice. The authors should show the phenotypes of osteoclast-specific arginase deficient mice.

Response:

We now included *Arg1*^{Tie2/+} conditional knockout animals, which selectively delete Arg1 in all hematopoietic cells including osteoclast precursors. Thereby we ensured active deletion of cellular Arg1 prior to induction of RANKL signaling cascades that induce osteoclastogenesis, mimicking the timing of recArg1 treatment. Deletion of cellular Arg1 in preosteoclasts exerted unremarkable effects on osteoclastogenesis (Fig. 3g). Further underscoring the importance of extracellular arginine degradation via recArg1 on osteoclastogenesis versus intracellular degradation mediated by cellular Arg1, we observed RANKL treatment downregulated cellular Arg1 both on protein and mRNA level (Fig. 3h, i, Supplementary Data Fig. 2a).

2. The authors should provide more detailed phenotypes that are typically analyzed in the field of bone biology in arginine deletion model (e.g. body size, body weight, serum calcium and phosphorus levels, whole-body X-ray picture, bone morphometric analysis and CT analysis).

Response:

We have now included the weights of all animals treated with recArg1. We could not observe striking weight differences and therefore did not observe any differences in body size upon recArg1 treatment (Fig. 2). The bone morphometric analysis of paws is included in the main figure (Fig. 2, osteoclast number and inflammation area). Unfortunately, in the course of these studies we did not perform CT analysis or X-ray, as these methods are not established in our institute. We strongly adhere to the three Rs (reduction, refinement and replacement) of animal welfare and therefore think that repeating these experiments for CT and X-ray analysis is unnecessary as the beneficial effects of recArg1 are observed using different

readouts (clinical scores, bone morphometric analysis coupled to histology, IVIS) in three different murine arthritis models.

3. Because the clinical data of recArg1 is now available, the authors should provide the human data.

Response:

Pegylated human arginase 1 (recArg1/BCT-100) was proven to be active and safe in patients suffering from hepatocellular carcinoma (Yau et al., Invest New Drugs, 2013; Yau et al., Invest New Drugs, 2015). These studies describe the pharmacokinetics and dynamics of recArg1 treatment in humans. To our knowledge, recArg1 treatment has not been used in the context of human rheumatoid arthritis treatment and thus, we cannot provide clinical data. Nonetheless, we have now evaluated arginine and arginase 1 levels in rheumatoid arthritis patients suffering from erosive or non-erosive rheumatoid arthritis. This new data strengthens the importance of arginine metabolism in the context of humans suffering from excessive bone degradation (Fig. 1a-c).

4. The authors should check the serum levels of OPG and RANKL of the mice in this study.

Response:

We thank the reviewer for this comment and have now included the RANKL/OPG ratios of animals used in this study (Supplementary Data Fig. 1h). The text now reads:
“Of note, in CIA, myeloid populations including osteoclast precursors were unaffected by arginine restriction, as was the systemic RANKL/OPG ratio in all models studied (Supplementary Fig. 1g-h), suggesting that decreased osteoclast numbers found in arthritis were due to differentiation of osteoclasts.”

Reviewer #2 (Remarks to the Author):

The manuscript by Brunner and colleagues examines the effect of arginine depletion on osteoclast polykaryon formation by a combination of in vitro (studies comprising transcriptomics, proteomics, metabolomics and OXPFOs) and in vivo models of arthritis. The results show that the use of FDA-approved BCT-100 inhibits polykaryon formation and reduces the severity of arthritis. The mechanisms through which this effect occurs are independent of mTOR activation, through a general blunting of RANKL signaling. The same conclusions are drawn for the IL-4 derived giant cells, even though only polykaryon formation and OXPFOs were tested in these cells. The general conclusion stresses the importance of nutrient availability for polykaryon formation, an energy demanding process that controls bone remodelling. The manuscript needs strengthening on two main grounds; the central hypothesis (1) and conclusions reached upon incomplete data (2 and 3).

Response:

We thank Reviewer #2 for his/her comments. We have performed several additional experiments that strengthen the conclusions and hypothesis of our manuscript. Please see below for detailed responses.

1. The rationale behind the choice of arginine depletion to study polykaryon formation and arthritis is not clear. Figure 4 shows that polykaryon formation is equally depending on leucine, isoleucine and phenylalanine and Extended Figure 7 shows that osteoclastogenesis does not strictly depend on essential aa (example of Gln). Importantly, metabolism of 13/20 aa controls osteoclastogenesis. For those 13 where commercial inhibitors like BCT-100 are available, the effect on the mice CIA should be explored. This is particularly important since there is no mechanism (signalling, transcriptional, metabolic or proteomic) which could be specifically attributed to arginine depletion in RANKL-dependent osteoclast formation.

Response:

Our rationale to study polykaryon formation and arthritis in the context of arginine depletion is that

- 1) We show in Figure 3b that arginine and proline metabolism are the most perturbed by RANKL.
- 2) Using transcriptomics and proteomics we further describe that arginine is critical for the RANKL mediated generation of TCA metabolites and transcription of associated enzymes (Fig. 4 and Fig. 5, Fig. S2 and Fig S5), i.e. arginine is crucial for RANKL signaling.
- 3) BCT-100 is an FDA approved drug, whose pharmacodynamics and pharmacokinetics have been published (Yau et al., Invest New Drugs, 2013; Yau et al., Invest New Drugs, 2015). It is considered to be active and safe in patients, yet its translational potential in arthritis has not been described.

Thus, we do not think the rationale is unclear. However, we have now added a section on the relevance of arginine metabolism in erosive arthritis patients, who suffer from enhanced osteoclast activity (Fig. 1). These data further underscore the potential relevance of manipulating arginine metabolism for arthritis treatment.

We evaluated the role of the other amino acids in osteoclastogenesis to demonstrate that following their lack, commonalities in metabolic blockage occurred (Fig. 7b and Supplementary Fig. 8). We further show that these blockages can be rescued by

supplementation of derivatives or precursors of these amino acids. These experiments therefore establish a common theme of polykaryon metabolic adaptation, where multi-nucleation is dependent on a nutrient-rich environment, broadening the scope of our manuscript.

In our manuscript we already extensively examined the role of extracellular arginine in three different arthritis models (Fig. 2). Given we have already performed several different models, together with the careful ethical considerations in performing yet additional animal models, we believe that examining if and how other amino acids modulate arthritis outcome is out of the scope of this study. Our study is not concerned with whether absence of other amino acids exerts potential beneficial effects in murine arthritis, but rather as the reviewer him/herself earlier points out is about “the importance of nutrient availability for polykaryon formation, an energy demanding process that controls bone remodeling”.

2. The metabolomics experiments are incomplete and a TCA rewiring cannot be concluded based on the presented data. ^{13}C arginine is used to measure cellular uptake (Figure 3g) but this does not inform on the specific contribution of labelled arginine to the TCA cycle. Targeted metabolomics with labelled arginine would allow an accurate contribution of this aa into TCA metabolites in osteoclasts. These experiments should be carried out in presence and absence of BCT-100. The TCA rewiring is concluded based on gene expression data in serine/purine synthesis pathways. Isotope tracing experiments will confirm if this is the case. Furthermore, the Seahorse experiments are performed 48h after RANKL stimulation, whereas metabolomics after 24h. This is problematic because the manuscript makes parallels between OXPFOs effect and metabolomics results, despite the fact that osteoclasts could be at different stages of their differentiation.

Response:

We thank reviewer #2 for this comment. We have now traced both $^{13}\text{C}_6$ arginine and $^{13}\text{C}_4$ aspartate during the first 24 hours of osteoclastogenesis and performed this experiment in the presence and absence of recArg1. We chose labelled aspartate in addition to arginine, as the carbon backbone of arginine remains in the urea cycle and cannot directly enter the TCA cycle. Aspartate further links the urea and TCA cycle via generation of argininosuccinate (Fig. 6a). Tracing arginine upon recArg1 treatment, we observed high intracellular labelling in products of the arginase 1 reaction. Concerning aspartate, we observed enhanced labelling of malate in comparison to aspartate input, which suggests a dysregulated TCA cycle. The text now reads:

“To distinguish between the possibilities noted above, we next performed ^{13}C -isotopologue labelling experiments. As the carbon backbone of arginine cannot serve as a direct source for TCA cycle intermediates, we supplied $^{13}\text{C}_6$ arginine in combination with $^{13}\text{C}_4$ aspartate in the first 24h of osteoclastogenesis and measured intracellular isotopologue distribution via LC-MS analysis. In this way, we could assess the contribution of each carbon source to both the urea and the TCA cycle simultaneously, as aspartate can be readily converted through argininosuccinate into fumarate via the TCA-Urea cycle shunt (Fig. 6a). $^{13}\text{C}_6$ arginine was readily taken up by both M-CSF and RANKL treated cells, as we detected 100-200 times more intracellular labelled arginine in relation to unlabelled arginine. In line with negligible increases in arginine transporter transcription (*Slc7a1*, *Slc7a2*) (Supplementary Fig. 2e), we observed no difference in $^{13}\text{C}_6$ labelled arginine uptake upon RANKL treatment post 24 hours. Both M-CSF and RANKL treated cells engaged in arginine recycling as indicated by *m*+5 labelling in both citrulline and arginine (Fig. 6b). Notably, as arginine was readily converted to ornithine and urea by recArg1 in the supernatant, we could barely detect labelled

intracellular $^{13}\text{C}_6$ arginine upon recArg1 treatment (Fig. 5f, Fig. 6b, Supplementary Fig. 7d). Relative to the low amounts of intracellular $^{13}\text{C}_6$ arginine, under arginine depletion, we detected high quantities of $^{13}\text{C}_5$ labelled ornithine and its downstream metabolites putrescine and spermidine, because the enhanced ornithine generated by recArg1 was further catabolized to form polyamines and came from the extracellular $^{13}\text{C}_6$ arginine supplied (Fig. 6c). Suggestive of uptake and/or synthesis from $m+5$ ornithine occurring under ^{13}C labelling, total intracellular amounts of the polyamines putrescine and spermidine were decreased upon recArg1 treatment in comparison to RANKL (Fig. 5f, Fig 6c, Supplementary Fig. 7c). RNA sequencing analysis demonstrated that similar to macrophages ornithine transcarbamylase (OTC) was not expressed in preosteoclasts²⁶ (data not shown), therefore ornithine cannot be converted to citrulline in these cells. Together, these data indicate that recArg1 treatment converts all extracellular arginine to ornithine and the polyamines as demonstrated by the lack of $m+5$ citrulline and $m+5$ arginine (Fig. 6b, Fig. 6c).”

and

“To prove an operating urea-TCA cycle shunt in preosteoclasts, we simultaneously performed a $^{13}\text{C}_4$ aspartate tracing. Suggesting the aspartate-argininosuccinate-fumarate shunt functioned in preosteoclasts independent of arginine presence, we observed equal $m+4$ fumarate labelling in in all conditions (Fig. 6f). Notably, as fumarate was $m+4$ labelled, we concluded the carbon source stemmed from argininosuccinate, as direct conversion from aspartate to oxaloacetate would have yielded $m+3$ fumarate (Fig. 6a). Consistent with arginine deprivation globally downregulating TCA cycle enzymes (Fig. 5g), we observed higher abundance of $m+4$ malate relative to $^{13}\text{C}_4$ aspartate input following recArg1 treatment compared to M-CSF and RANKL control conditions (Fig. 6f). Although arginine and its catabolic products cannot directly enter the TCA cycle, the metabolic tracing data utilizing fully-labelled aspartate strengthen the notion that arginine presence is required for a functioning TCA cycle.”

Regarding the different timepoints used for Seahorse experiments (48h) and metabolomics (24h), we have now performed Seahorse at 24h post RANKL and recArg1 treatment. These data show that RANKL does not induce OXPHOS at 24 hours i.e. the increases in RANKL-induced TCA metabolites precede the changes in cellular respiration (Supplementary Fig. 6e). The text now reads:

“Oxygen consumption rates 24h post RANKL treatment were unchanged compared to M-CSF, indicating RANKL induced changes in TCA metabolites preceded the observed boost in OXPHOS (Supplementary Fig. 7e).”

3. The IL-4 derived giant cells data are incomplete if one would like to conclude similarities with osteoclasts in the abstract and title. Tracing experiments with labelled arginine will be necessary to claim that a similar metabolic adaptation occurs. The manuscript claims that arginine depletion specifically abolishes RANKL signaling in osteoclasts. If same is true in IL-4 signaling, then this treatment has pleiotropic effects and is not pathway-specific as suggested.

Response:

We do not believe that metabolic tracing in IL-4 MGCs is necessary to claim metabolic similarities between osteoclasts and IL-4 MGCs. Our data indicate that akin to osteoclasts arginine deprivation blocks IL-4 induction of MGCs. We further show that attenuated MGC formation caused by scarcity in arginine can be compensated by citrulline and argininosuccinate in both models, indicating similar cellular adaptation. Further, both cell types displayed clear parallels in arginine dependent oxidative metabolism. To strengthen this

notion, we now performed experiments showing that IL-4 MGC and osteoclast formation triggered similar metabolic enzyme expression reliant on arginine (Fig. 8c).

Although, we agree that RANKL and IL-4 trigger different signaling cascades and that these cell types are not the same, the aforementioned experiments indicate energy dependent processes ensure giant cell formation and arginine is critical for this. We thus believe the title “Metabolic Adaptation Controls Multinuclear Giant Cell Formation” adequately reflects the data outlined in our manuscript. Nonetheless, we agree that the language indicating a conserved mechanism in the abstract was too bold and have now changed this to read: “Arginine scarcity also dampened generation of another form of MGCs, IL-4 induced polykaryons.”

Minor

1. There are many shortcuts in reaching conclusions. To discriminate between inflammation and osteoclast driven damage in arthritis models, requires genetic approaches (Ctsk-driven conditional KO). Mentioning an osteoclast-specific effect of the drug is an over-statement and should be avoided.

Response:

Thank you for this suggestion, but as our study focuses on preosteoclasts and osteoclast differentiation, we do not think a Ctsk-driven conditional knockout (deletion after induction of differentiation) would be helpful. However, we have included a cellular *Arg1* conditional knockout, which selectively deletes *Arg1* in all hematopoietic cells including osteoclast precursors. Cellular *Arg1* deletion exerted unremarkable effects on osteoclastogenesis (Fig. 3g). Notably, we observed downregulation of cellular *Arg1* upon RANKL treatment on both the protein and mRNA level, further underscoring the importance of extracellular arginine degradation via recArg1 versus intracellular degradation mediated by cellular *Arg1* on osteoclastogenesis (Fig. 3h,i, Supplementary Data Fig. 2a).

2. Line 186-206. The paragraph has speculative statements. There is no data to support a transient pre-osteoclast state.

Response:

We do not explicitly say that there is a transient preosteoclast state. We instead infer that arginine deprivation results in a potential “transient reversible block in osteoclastogenesis”. Data that supports this claim is provided in Fig. 4b and Supplementary Data Fig. 3c,d. The data in Fig. 4b suggests a transcriptional signature upon arginine deprivation that is neither associated with M-CSF nor with M-CSF+RANKL treatment, but rather is somewhere in-between. We further show that arginine deprivation induced blocks in osteoclastogenesis can be reversed and osteoclast formation is restored after arginine re-supplementation post recArg1 treatment (Supplementary Data Fig. 3c,d).

3. The text is difficult to follow at times (e.g. page 8, line 164, the sentence starting with ‘subtle effects’ needs revising)

Response:

We changed the text to:

“Using a metabolomic approach, we observed RANKL dependent induction of key intracellular TCA cycle intermediates (e.g. malate, fumarate, succinate, α -ketoglutarate) 24h

post stimulation and only subtle changes in intracellular AAs (e.g. arginine, glycine, alanine) (Fig. 5f, Supplementary Fig. 7c).”

4. Alpha-ketoglutarate is misspelled on numerous occasions, including in figures (Figure 3)

Response:

We apologize for this mistake. We have corrected all misspellings of alpha-ketoglutarate including in Fig. 5 (old Fig. 3).

5. Line 238-241. The general conclusion is highly speculative and does not reflect the data presented in the manuscript.

Response:

Given we have expanded the manuscript, we have changed this part of the discussion.

6. Given the major differences in urea cycle between humans and mice (absence of nitric oxide synthase induction in inflammatory human mononuclear phagocytes), it is not accurate to claim in the title that ‘metabolic adaptation controls multinucleated giant cell formation’. The manuscript shows data on the metabolism of murine osteoclast formation and it is unclear if the same applies to the IL-4-induced polykaryons and to human osteoclasts and giant cells.

Response:

As mentioned earlier, we agree that RANKL and IL-4 trigger different signaling cascades and that these cell types are not identical. Nonetheless, our data indicate arginine is critical for energy dependent processes that ensure both types of giant cell form. Additionally, we point out that precursors or intermediates of arginine rescue both types of MGC formation in arginine absence. We further demonstrate similarities in adaptative responses to lack of other amino acids than arginine in osteoclastogenesis. We also show that arginine scarcity blocks human osteoclast formation (Fig. 3c). We thus believe the title “Metabolic Adaptation Controls Multinuclear Giant Cell Formation” adequately reflects the data described.

Reviewer #3 (Remarks to the Author):

Brunner et al demonstrate that the administration of recombinant Arginase-1 (recArg1) inhibits osteoclastogenesis and improves disease outcome in several arthritis mouse models. They further show that reducing systemic arginine levels does not globally affect the immune system but quite specifically affects the formation of osteoclasts.

In a series of in vitro experiments, the authors demonstrate that RANKL-mediated osteoclast differentiation was associated with a strong increase in the arginine biosynthesis pathway. Consistent with an important role for arginine metabolism in osteoclast differentiation, the addition of recArg1 to the culture medium inhibited the differentiation of osteoclasts, whereas recArg1 only exerted minor effects on macrophage and DC differentiation.

Based on a detailed transcriptomics analysis, the authors inferred that recArg1 inhibited osteoclast differentiation through depletion of arginine rather than through the increased generation of downstream metabolites such as urea and ornithine. The authors further show that arginine depletion by recArg1 inhibits RANKL-induced effects on preosteoclasts but does not globally reduce translation, which one would perhaps expect given that arginine is required for protein synthesis.

In the second part of the manuscript, Brunner et al report that the absence of arginine hampered osteoclast differentiation independent of mTOR signaling components or putative arginine sensors. Through a series of experiments addressing the cellular metabolism of preosteoclasts the authors conclude that arginine catabolism sustains OXPHOS. In addition, the authors show that precursors of arginine or downstream metabolites of other essential amino acids can compensate for the absence of the respective amino acid.

Overall, this is an interesting study and the finding that administration of recArg1 attenuates arthritis has translational potential. The figures are clear and well presented. However, the manuscript is sometimes difficult to understand probably because the format is rather short. Some of the experiments that were performed to study metabolism in preosteoclasts need further clarification (detailed below).

Response:

We thank reviewer #3 for stating that our study is interesting and that recArg1 might have translational potential in arthritis. In the aim of aiding understanding, we have now expanded on the length of the manuscript. We have further made clearer that “recArg1” conditions refer to “M-CSF+RANKL+recArg1” and have dampened down on the language, which suggested direct TCA fueling through amino acid catabolism. Please see below for responses to specific comments.

Specific comments:

1.) Some parts of the manuscript and figures require a more detailed description:

- For example, the authors state: “RANKL-induced bursts in preosteoclast SRC were counteracted by recArg1”. From this one would conclude that the third condition in Fig 3c is “RANKL + recArg1” but it is just labeled with recArg1. It would need clarification if only recArg1 was added. All panels throughout figure 3 were labeled this way.

Response:

We have now changed the text of Fig. 3a to read: “To investigate the molecular and cellular mechanisms associated with arginine restriction in bone disease, we next dissected osteoclastogenesis induced with M-CSF/RANKL in the presence or absence of recArg1 using different systems-type approaches. For simplicity, we refer to M-CSF+RANKL treatment as “RANKL” and M-CSF+RANKL+recArg1 treatment as “recArg1”, unless otherwise indicated (Fig. 3a).”

We have further included a box containing the labelling used throughout Fig. 5 (old Fig. 3), to clarify that recArg1 indicates M-CSF+RANKL+recArg1.

- There are several sentences that are rather short and difficult to understand. For example, the authors state “RANKL induced glycolysis irrespective of glucose uptake and arginine presence” (Line 161). Only after studying the figure it became clear to me that RANKL enhances the glycolytic rate in preosteoclasts without markedly increasing glucose uptake...In this reviewer’s opinion, several sentences should be rewritten to help the reader easily understand the results. The figures, however, are very nice and easy to understand.

Response:

We have now changed the text to:

“Moreover, versus the M-CSF control, RANKL markedly induced preosteoclast glycolytic rates, independent of changes in glucose uptake (Fig. 5d-e). Elevated preosteoclast glycolysis under RANKL treatment was independent of arginine as no decrease was observed with recArg1 addition (M-CSF+RANKL+recArg1) (Fig. 5d-e).”

- Another example is the following sentence:

“Subtle effects were observed on other AAs (e.g. glycine, alanine), suggesting specific arginine requirements for RANKL signalling, corroborated by enhanced ¹³C labelled arginine intracellular build-up and extracellular arginine uptake from the supernatant, disassociated from arginine transporter transcription (Slc7a1, Slc7a2)”

How does enhanced arginine uptake corroborate the previous observation?

Response:

We have changed this sentence and have further performed metabolic tracing experiments with labelled ¹³C₆ arginine to clarify the impact of recArg1 on intracellular arginine metabolism (Fig. 6).

- The authors state “RecArg1 caused intracellular arginine build-up...(Figure 3f, h)”. Looking at Fig 3f. it seems that both conditions labeled as “RANKL” and “RecArg1” increase intracellular arginine levels to the same extent. Why is only RecArg1 mentioned and is this condition RANKL + recArg1 or only recArg1?

Response:

As mentioned earlier, we have further elaborated on the fate of intracellular arginine using isotopologue labelling in Fig. 6. We further included a box containing the labelling used throughout Fig. 5, making it clearer that recArg1 always refers to M-CSF+RANKL+recArg1.

2.) In Figure 3f, perhaps the most striking differences between the two conditions “RANKL” and “recArg1” were in putrescine and spermidine pointing to polyamine metabolism. Given that the authors report that recArg1 inhibits osteoclast differentiation, it might be interesting

to address whether a decrease in polyamines - which are known to affect proliferation and differentiation - plays a role in osteoclastogenesis.

Response:

We thank reviewer #3 for this suggestion. Polyamines have already been investigated in the context of osteoclastogenesis and their extracellular addition is described to negatively impact it (Yeon et al., Amino Acids, 2014). However, we performed related experiments and have included them in the manuscript:

“It was previously shown that extracellular polyamines negatively regulate osteoclastogenesis²⁷. However, although recArg1 led to a shift in putrescine and spermidine amounts, their addition at excess (500 μ M putrescine and 10 μ M spermidine) during osteoclastogenesis caused an approximately 30% and 75% decrease in TRAP positive cells respectively. Thus, although accumulation of polyamines upon recArg1 might negatively impact osteoclastogenesis, their increased synthesis cannot fully account for the complete absence of TRAP positive cells observed under arginine deprivation (Fig. 6e). As we wanted to test if inhibitory effects of recArg1 treatment on osteoclastogenesis were mediated by decreased intracellular polyamine synthesis (Fig. 5f, Supplementary Fig. 7c), we incubated cells with alpha-difluoromethylornithine (DFMO), thereby blocking conversion of ornithine to putrescine. Despite DFMO reducing osteoclastogenesis, multinucleated mature osteoclasts were still observed (Fig. 6e). Overall, these data suggest an importance for polyamines in the context of osteoclastogenesis. However, the effects of recArg1 cannot be solely accounted by deregulated polyamine metabolism.”

3.) Based on transcriptomics analyses, the authors inferred that recArg1 inhibited osteoclast differentiation through depletion of arginine rather than through the increased generation of downstream metabolites such as urea and ornithine. This could be easily validated by adding urea or ornithine to the culture medium.

Response:

We acknowledged this suggestion and have performed the suggested experiment. The text relating to this now reads:

“Excess of extracellular ornithine and urea did not have any consequences on osteoclastogenesis, proving that the recArg1 instigated blocks in osteoclastogenesis were not due to increases in these metabolites (Fig. 6d).”

4.) The authors mention that the urea cycle and the TCA are linked through fumarate and conclude that arginine catabolism sustains RANKL-dependent OXPHOS. This might leave the reader somehow with the impression that metabolites of the urea cycle are shunted into the TCA cycle to enhance OXPHOS but there is no evidence for this and would require metabolic tracing (flux) experiments. In this reviewer’s opinion, this should be either experimentally addressed or discussed in a separate section.

Response:

We appreciate this comment. As mentioned earlier, we have now performed metabolic tracing experiments and added them to the manuscript (Fig. 6). Given that this data points towards a dysregulated TCA cycle, we have further adjusted the language and discussion of the manuscript.

5.) The idea that catabolic products directly fuel the TCA cycle is taken up again in the last section where the authors conclude: “Importantly, these data suggest a common theme by

which osteoclasts dynamically metabolically adapt to the absence of environmental AAs by utilizing their respective TCA fuelling intermediates". This conclusion is based on the finding that arginosuccinate or citrulline can compensate for the absence of arginine. However, this is expected as those are the precursors of arginine and can be readily converted into arginine by the cells. As such, this does not provide evidence that citrulline and arginosuccinate sustain the TCA cycle.

Response:

We changed this sentence. The text now reads:

“Therefore, these data suggest that osteoclasts can dynamically adapt their metabolism to the absence of environmental AAs by utilizing their respective precursors or intermediates for successful osteoclastogenesis.”

As mentioned above, we have changed the discussion and adjusted the language based on our newly added experiments.

The author’s conclusion that amino acid derivatives fuel the TCA cycle is further based on the observation that alpha-ketoisocaproate can compensate for the absence of leucine. The conversion of leucine into alpha-ketoisocaproate is catalyzed by BCAT1/2, yet this is a reversible reaction. Hence, if alpha-ketoisocaproate is added to the culture medium it is readily converted to leucine by the cells. If amino acids were only required to fuel the TCA cycle, then the addition of TCA cycle metabolites, such as a-KG should compensate for the absence of essential amino acids.

The described experiments recapitulate known biochemistry but do not provide enough evidence to conclude that “scarcity in any single essential AAs can be compensated by its derivatives or precursors to sustain oxidative metabolism”. (line 239)

Response:

We agree that alpha-ketoisocaproate can be converted back to leucine. We have dampened down on the language that amino acids directly fuel the TCA cycle, as highlighted in the text: “TCA cycle intermediates (e.g. fumarate and α -ketoglutarate) were not able to rescue arginine absence (data not shown), suggesting that arginine scarcity could not be compensated by direct TCA cycle refuelling but only by AA derivatives.”

We further changed this sentence and the related discussion to:

“Thus, osteoclasts most likely couple extracellular AA amounts to their developmental program, as scarcity in single essential AAs could be compensated by osteoclast intrinsic re-synthesis through supplementation of its derivatives or precursors. While here we established TCA cycle dysregulation upon arginine withdrawal, we cannot be certain if this is true for the remaining osteoclastogenesis essential amino acids identified in this study.”

Reviewers' Comments:

Reviewer #1:

Remarks to the Author:

The authors have adequately addressed my concerns and as a consequence the findings have been significantly strengthened and extended.

Reviewer #2:

Remarks to the Author:

The flaws of the manuscript were summarized as 3 major points.

1. The authors rebutted the rationale behind the choice of arginine to study osteoclast formation and added Figure 1 as new data. The current version of the manuscript argues against the importance of arginine in osteoclast driven RA as the newly added data in erosive and non-erosive RA patients show non-statistically significant serum arginine levels between the two (Figure 1b). Arginase 1 is significant between the two subgroups but the correlation with erosion score is weak. Based on these new results, the choice of arginine to study murine osteoclastogenesis and metabolism is questionable. An appropriate study design would have used the RA setup for the detection of all amino acids and focus on the ones that show the most significant changes in view of human osteoclastogenesis RANKL transcriptomics and/or metabolomics data. Furthermore, the statement in the abstract that and the rebuttal that 'the polykaryon formation requires a metabolic adaptation because of restriction with multiple amino acids' is an erroneous interpretation of the data. The words 'metabolic adaptation' refer to experiments measuring metabolic fluxes in a cell type whereby a given metabolic pathway is preferentially used over others. None of the findings presented in Figure 7 and Supplementary Figure 8 show data on metabolic adaptation with other aa – only TRAP staining is shown which reflects osteoclast formation.

2. The authors have performed ¹³C arginine and ¹³C aspartate tracing experiments in order to dissect the relative contribution of each aa to urea and TCA cycle, respectively. However these experiments complicate further the main message as they show that recArg1 converts the extracellular arginine to ornithine and polyamines that inhibits osteoclastogenesis and explains a significant fraction of the BCA-100 effect on osteoclastogenesis. Notably, the tracing experiments conclude that labelled arginine and its downstream metabolites do not enter the TCA cycle. The aspartate labelling experiments bring additional questions: Why recArg1 treatment causes an increase in m+4 malate but not m+4 fumarate? This suggests that arginine depletion modifies the aspartate-as-fumarate shunt. Also, not all the isotopologues are shown – what happened to unlabelled (m+0) metabolites? In summary, arginine deprivation shows transcriptomics changes in TCA cycle enzymes (Figure 5g) but this is due to indirect effects, independent of arginine metabolism that remain unexplored. Based on these results, the abstract cannot claim a 'dysregulated TCA cycle' as arginine does not enter the TCA cycle. This conclusion was reached solely on transcriptomics data, which, on its own, is not sufficient evidence.

3. A 'metabolic adaptation' refers to a given cell type in a given specie, under a specific stimulus that shows metabolic fluxes that differ from the basal state. Claiming that RANKL stimulation causes similar metabolic changes as IL-4 without measurement of any intracellular metabolite is factually wrong and gene expression data (Figure 8c) does not prove that. As a result, the title of the manuscript does not reflect the data presented in the manuscript. The abstract and the main text also claim incorrectly the metabolic adaptation of IL-4-derived polykaryons.

Reviewer #3:

Remarks to the Author:

The authors have provided additional data, improved the manuscript and have addressed all my concerns.

Responses to Reviewer #2

We would like to thank reviewers #2 for his/her thoughtful comments and suggestions. We have addressed each point in full and made relevant changes to the text that we think have significantly improved the manuscript's quality. Specifically in relation to each comment:

1. The authors rebutted the rationale behind the choice of arginine to study osteoclast formation and added Figure 1 as new data. The current version of the manuscript argues against the importance of arginine in osteoclast driven RA as the newly added data in erosive and non-erosive RA patients show non-statistically significant serum arginine levels between the two (Figure 1b). Arginase 1 is significant between the two subgroups but the correlation with erosion score is weak. Based on these new results, the choice of arginine to study murine osteoclastogenesis and metabolism is questionable. An appropriate study design would have used the RA setup for the detection of all amino acids and focus on the ones that show the most significant changes in view of human osteoclastogenesis RANKL transcriptomics and/or metabolomics data. Furthermore, the statement in the abstract that and the rebuttal that 'the polykaryon formation requires a metabolic adaptation because of restriction with multiple amino acids' is an erroneous interpretation of the data. The words 'metabolic adaptation' refer to experiments measuring metabolic fluxes in a cell type whereby a given metabolic pathway is preferentially used over others. None of the findings presented in Figure 7 and Supplementary Figure 8 show data on metabolic adaptation with other aa – only TRAP staining is shown which reflects osteoclast formation.

Response:

To sustain their proliferation cancer cells can exhibit metabolic adaptation to changed extracellular nutrients. However, compensation cannot occur for every nutrient as enzymes that deplete specific amino acids e.g. arginine, can be utilized to limit auxotrophic cancer cell growth. Although pegylated human recombinant arginase 1 (BCT-100, recArg1) has been used in hepatocellular carcinoma (Yau et al., Invest New Drugs, 2013; Yau et al., Invest New Drugs, 2015), it has not been used in rheumatoid arthritis. Thus, as highlighted in the introduction, the effects of arginine availability on other disease settings are unknown and we therefore began this study as we reasoned arginine availability might regulate murine arthritis. Given these experiments demonstrated recArg1 exerted its beneficial effects likely by dampening osteoclastogenesis, we believe that the choice to study arginine in osteoclast formation is clear. This is further supported by our unbiased analysis of the most pronounced induced metabolic processes by RANKL, which revealed that processes related to arginine rank on top (New Fig. 2b). To further clarify the rationale behind systemically manipulating arginine in the context of arthritis, we have expanded on the introduction.

A manuscript is not necessarily written in the order of performed experiments. During the first round of review and on another reviewer's request, we retrospectively evaluated arginine and arginase 1 levels in rheumatoid arthritis patient cohorts. We believe that determining which amino acids are most significantly changed in these patients and examining how their lack influences RANKL transcriptomics and/or metabolomics would not provide a sound starting point for our manuscript, as this would greatly expand the existing manuscript, likely moving it into new directions.

We consider the weak (likely explained by limitations in patient sample size), albeit statistically significant correlation shown in Fig. 1b (new Fig. 1f) does not argue against the

importance of the arginine pathway in erosive rheumatoid arthritis, but rather supports our murine data. To better reflect this and the order of experiments, we have now shifted the human data to the end of New Fig. 1.

As mentioned in our first response to reviewer #2, we evaluated the role of other amino acids in osteoclastogenesis to show that following their lack, osteoclastogenesis was blocked and importantly these blocks could be rescued by supplementation of derivatives or precursors of these amino acids. Although these experiments support adaptive responses during osteoclastogenesis, we agree with the reviewer that the data in Fig. 6 (old Fig. 7) and Supplementary Fig. 8 do not yet define a metabolic mechanism per se, which likely will require much further work by diverse research groups. We thank reviewer #2 for this fair comment and in this regard, we have now modified the corresponding statement in the abstract, focusing more on arginine and we write “Strikingly, in the absence of extracellular arginine, both cell types displayed flexibility as their formation could be restored with select arginine precursors.”

We would like to point out that the transcriptomics, proteomics, metabolomics and Seahorse data (New Fig. 3, Fig. 4, Supplementary Fig. 6) collectively indicate arginine presence is required for the induction of oxidative metabolism that we in turn show is critical for the formation of TRAP positive, multinucleated osteoclasts. Thus, it is plausible that the lack of osteoclasts upon deprivation of these other amino acids impinges upon blocks in OXPPOS. Indeed, we further elaborate on this topic in the discussion, stating that “it is plausible that similar mechanisms of energy shifts under amino acid starvation are in place as observed under arginine withdrawal”.

2. The authors have performed ^{13}C arginine and ^{13}C aspartate tracing experiments in order to dissect the relative contribution of each aa to urea and TCA cycle, respectively. However these experiments complicate further the main message as they show that recArg1 converts the extracellular arginine to ornithine and polyamines that inhibits osteoclastogenesis and explains a significant fraction of the BCA-100 effect on osteoclastogenesis. Notably, the tracing experiments conclude that labelled arginine and its downstream metabolites do not enter the TCA cycle. The aspartate labelling experiments bring additional questions: Why recArg1 treatment causes an increase in m+4 malate but not m+4 fumarate? This suggests that arginine depletion modifies the aspartate-as-fumarate shunt. Also, not all the isotopologues are shown – what happened to unlabelled (m+0) metabolites? In summary, arginine deprivation shows transcriptomics changes in TCA cycle enzymes (Figure 5g) but this is due to indirect effects, independent of arginine metabolism that remain unexplored. Based on these results, the abstract cannot claim a ‘dysregulated TCA cycle’ as arginine does not enter the TCA cycle. This conclusion was reached solely on transcriptomics data, which, on its own, is not sufficient evidence.

Response:

At reviewer #2’s and reviewer #3’s previous suggestions, we performed $^{13}\text{C}_6$ arginine tracing to examine contributions of arginine presence to the RANKL mediated TCA cycle. The reviewer is correct that all labelled extracellularly supplied arginine is converted to ornithine by recArg1. The resulting ornithine can be further converted to the polyamines by ornithine decarboxylase. However, we do not consider that the generation of these downstream metabolites complicates the main message nor explains all effects of recArg1 on osteoclastogenesis.

Indeed, excess of extracellularly supplied ornithine and urea has no inhibitory effects on osteoclastogenesis (New Fig. 5d). Further, excess polyamines, unlike recArg1, dampen but do not completely inhibit osteoclastogenesis (New Fig. 5e). Preosteoclasts cultured in arginine free medium or treated with recArg1 display very similar transcriptional profiles (New Fig. 3c) and akin to recArg1, medium devoid of arginine blocks osteoclastogenesis (New Fig. 6a, Supplementary Fig. 8). Given the arginase 1 reaction is not enhanced in arginine free media that lacks recArg1, we argue that the generation of downstream metabolites by recArg1 cannot solely account for its effects on osteoclastogenesis.

As the carbon-atoms of arginine cannot directly enter the TCA cycle, we combined $^{13}\text{C}_6$ arginine tracing with $^{13}\text{C}_4$ aspartate tracing, as aspartate can enter the TCA cycle via argininosuccinate and is further converted to fumarate. Aspartate therefore links the urea and the TCA cycle via argininosuccinate (New Fig. 5a). We performed this experiment to show an operating urea-TCA cycle shunt in preosteoclasts and to examine the effects of recArg1 herein. Given turnover of malate and fumarate is fast and we supplied excess labelled aspartate for 24 h it can be expected that the largest portion of metabolite pool gets labelled, therefore we could not detect unlabeled m+0 fumarate or m+0 malate in many samples (metabolites detected in 1/12 or 4/12 samples respectively).

We do not know exactly why recArg1 specifically leads to significant increases in m+4 malate but not m+4 fumarate. However, the biochemical reaction that converts fumarate to malate has a negative Gibbs free energy, indicating that it can occur spontaneously. There might be some biochemical advantages to convert excess fumarate pool to malate e.g. to provide enough substrate for the downstream NADH dependent reaction.

The reviewer correctly mentioned that m+4 malate accumulation is consistent with the downregulation of TCA cycle enzymes (we mentioned this in the manuscript), which further supports the notion that arginine presence is required for a functional TCA cycle that operates in both M-CSF and RANKL conditions.

The fact that arginine does not directly enter the TCA cycle does not exclude indirect TCA cycle dysregulation induced by environmental arginine absence. As mentioned earlier, transcriptomics, proteomics, network analysis, metabolomics and Seahorse data (New Fig. 3, Fig. 4, Supplementary Fig. 6) all indicate oxidative metabolism is crucially linked to arginine presence during RANKL signaling. Thus, m+4 malate accumulation that specifically occurs under recArg1 treatment further supports the occurrence of a changed TCA cycle versus M-CSF and RANKL conditions. Nonetheless, we have removed the word “dysregulation” in the abstract and replaced it with “impaired tricarboxylic acid (TCA) cycle function and metabolite induction” and hope this wording more adequately reflects the totality of the data.

3. A ‘metabolic adaptation’ refers to a given cell type in a given specie, under a specific stimulus that shows metabolic fluxes that differ from the basal state. Claiming that RANKL stimulation causes similar metabolic changes as IL-4 without measurement of any intracellular metabolite is factually wrong and gene expression data (Figure 8c) does not prove that. As a result, the title of the manuscript does not reflect the data presented in the manuscript. The abstract and the main text also claim incorrectly the metabolic adaptation of IL-4-derived polykaryons.

Response:

We thank the reviewer for giving his/her opinion on the definition of “metabolic adaptation” and acknowledge that metabolic adaptation might relate to “metabolic fluxes in a given cell type and species under a specific stimulus that differ from the basal state”. However, recent studies suggest that in the field of immunology, it is also described as the ability of resident immune cells to dynamically adjust their metabolism according to available extracellular energy sources in various tissue niches (Reviewed e.g. in Pearce et al., 2013, Immunity; Caputa et al., 2019, Nature Immunology). Thus, we and others believe that the concept of adaptation intrinsically involves metabolic flexibility following changes in environmental cues.

As highlighted above, transcriptomics, proteomics, network analysis, metabolomics and Seahorse data (New Fig. 3, Fig. 4, Supplementary Fig. 6) provide 5 lines of evidence that point towards oxidative metabolism being crucial for osteoclast differentiation. We demonstrate that arginine scarcity during osteoclast differentiation is associated with metabolic flexibility as it results in evident alterations of the TCA cycle, without inducing cell death. Preosteoclasts therefore alter their metabolism in response to arginine scarcity: this is a clear fact from our data. Demonstrating flexibility, arginine scarcity induced blocks in osteoclastogenesis can be restored by re-supplementation of arginine itself or by citrulline and argininosuccinate.

We apologize for the confusion in our previous response to reviewer #2. We did not mean to convey the message that RANKL and IL-4 induced MGCs share an identical metabolic program and agree that the data in New Fig. 7c does not reflect this. We have removed the sentence in the results stating “The observed identical regulation of these enzymes in IL-4 induced MGCs suggested metabolic commonalities between both types of MGCs.”.

However, as mentioned in our first response, our data indicate:

- environmental arginine is necessary for both types of MGC formation (New Fig. 2, Fig. 7)
- extracellular arginine is critical for OXPHOS in both types of MGCs (New Fig. 4c, Fig. 7f)
- select metabolic marker genes are identically regulated under extracellular arginine absence in both types of MGCs (New Fig. 7c)
- precursors of arginine can compensate for impairments in formation of both types of MGCs under environmental arginine scarcity (New Fig. 6a, Fig. 7d)
- in IL-4 induced polykaryons these precursors restore OXPHOS (New Fig. 7f)

To conclude, adaptive responses associated with metabolic flexibility upon arginine scarcity in both types of MGCs are apparent. However, we have modified the abstract to “Strikingly, in the absence of extracellular arginine, both cell types displayed flexibility as their formation could be restored with select arginine precursors”. To be more specific in relation to the findings described in the modified abstract, we further changed the title to “Environmental Arginine Controls Multinuclear Giant Cell Metabolism and Formation”. We have as well changed the ending of the manuscript to read: “Together, our data establish how availability of environmental amino acids, especially arginine, control polykaryon developmental programs and metabolism. They further imply therapeutic strategies for arginine depletion in MGC mediated diseases.”

Reviewers' Comments:

Reviewer #2:

Remarks to the Author:

The manuscript demonstrates that preosteoclasts alter their metabolism in response to arginine scarcity without providing the primary mechanisms. New Fig. 3, Fig. 4, Supplementary Fig. 6 show metabolic alterations due to a general inhibition of osteoclastogenesis, a known and widely described concept that could have been obtained with any blocker of this process. This lack of mechanistic insights makes the rationale behind the choice of arginine problematic, especially given the known discrepancy between the murine and human NOS-arginine pathways.

3rd revision

Responses to Reviewer #2

In general:

1. In the first revision (NCOMMS-19-09003A), reviewer #2 had various concerns and requested many experiments, several of which we performed: e.g. $^{13}\text{C}_6$ arginine tracing in osteoclasts in the presence and absence of recArg1. We additionally responded to all concerns raised.
2. In our second rebuttal to reviewer #2 (NCOMMS-19-09003B), we also addressed all newly raised concerns:
 - a. Explaining our choice behind studying extracellular arginine availability.
 - b. Why the tracing experiments do not complicate our main message.
 - c. Why we used the term metabolic adaptation for both types of MGCs.
 - d. To find a consensus with reviewer #2's concerns, we modified the text, abstract and title.
3. Despite these efforts on which reviewer #2 did not comment, he/she has now raised additional and unsubstantiated claims regarding both the rationale and mechanism(s) whereby arginine scarcity impacts osteoclastogenesis.

Response to reviewer #2:

The manuscript demonstrates that preosteoclasts alter their metabolism in response to arginine scarcity without providing the primary mechanisms. New Fig. 3, Fig. 4, Supplementary Fig. 6 show metabolic alterations due to a general inhibition of osteoclastogenesis, a known and widely described concept that could have been obtained with any blocker of this process. This lack of mechanistic insights makes the rationale behind the choice of arginine problematic, especially given the known discrepancy between the murine and human NOS-arginine pathways.

1. The reviewer is mistaken that our study lacks mechanistic insights linked to metabolic alterations upon arginine starvation occurring due to a general inhibition of osteoclastogenesis:
 - a. While osteoclastogenesis can be attenuated in many ways e.g. through inhibition of M-CSF or RANKL signalling^{1,2} or epigenetics³, there is no unifying theme. While all osteoclastogenesis inhibitors may or may not impact cellular metabolism, here we demonstrate for the first time the role of the extracellular arginine in this process.
 - b. Arginine is not a drug but an amino acid, whose lack forces preosteoclasts to change their metabolism, that is most importantly distinct from M-CSF stimulated cells (Fig. 3b, Fig. 4d,f-g, Fig. 5b,c,f). Effects of arginine deprivation on preosteoclast oxidative

metabolism are not due to a general shutdown of osteoclastogenesis, but rather reflective of metabolic quiescence.

- c. The effects of arginine starvation are reversible and can be compensated by urea cycle metabolites (Fig. 6a, Supplementary Fig. 3c-d).
 - d. Arginine affects osteoclastogenesis in a manner distinct to T cells^{4, 5}, therefore our study provides novel insights on how arginine is required for cellular metabolism during giant cell differentiation.
2. The rationale behind the choice of arginine to study osteoclastogenesis is clear:
 - a. Arginine metabolism is the most perturbed metabolic pathway in RANKL induced osteoclastogenesis as evidenced by unbiased RNA-sequencing (Fig. 2b).
 - b. In fact, the comprehensive systems biology approaches outlined in Fig. 3, Fig. 4 and Supplementary Fig. 6 show for the first time that extracellular arginine presence is required to sustain oxidative metabolism in osteoclasts. The tracing experiment (Fig. 5) further supports this notion. This is novel and not a widely described concept.
 - c. We have a clinical grade drug available that specifically depletes serum arginine, whose therapeutic potential in RA has not been described. Here we importantly show a benefit of depleting extracellular arginine in three models of murine arthritis (Fig. 1). Given recArg1 also blocks IL-4 induced MGC formation, our data suggest its beneficial effects may also apply to other MGC driven diseases.
 3. As we focus on how the lack of extracellular arginine impacts giant cell formation, the disputed discrepancies⁶ between murine and human arginase and NOS expression or their cellular source *in vivo* are of no relevance to our study. Indeed:
 - a. Cellular arginase 1 is downregulated by RANKL and its deletion has no impact on osteoclastogenesis (Fig. 2g-i).
 - b. We show that human osteoclastogenesis is as well dependent on extracellular arginine (Fig. 2c).

References:

1. Liao, H.J., Tsai, H.F., Wu, C.S., Chyuan, I.T. & Hsu, P.N. TRAIL inhibits RANK signaling and suppresses osteoclast activation via inhibiting lipid raft assembly and TRAF6 recruitment. *Cell Death Dis* **10**, 77 (2019).
2. Chang, E.J. *et al.* Hyaluronan inhibits osteoclast differentiation via Toll-like receptor 4. *J Cell Sci* **120**, 166-176 (2007).
3. Park-Min, K.H. *et al.* Inhibition of osteoclastogenesis and inflammatory bone resorption by targeting BET proteins and epigenetic regulation. *Nat Commun* **5**, 5418 (2014).
4. Geiger, R. *et al.* L-Arginine Modulates T Cell Metabolism and Enhances Survival and Antitumor Activity. *Cell* **167**, 829-842 e813 (2016).
5. Fletcher, M. *et al.* L-Arginine depletion blunts antitumor T-cell responses by inducing myeloid-derived suppressor cells. *Cancer Res* **75**, 275-283 (2015).
6. Caldwell, R.W., Rodriguez, P.C., Toque, H.A., Narayanan, S.P. & Caldwell, R.B. Arginase: A Multifaceted Enzyme Important in Health and Disease. *Physiol Rev* **98**, 641-665 (2018).